



# Steepening of magnetosonic waves in the inner coma of comet 67P/Churyumov-Gerasimenko

Katharina Ostaszewski[1], Karl-Heinz Glassmeier[1, 2], Charlotte Goetz[3], Philip Heinisch[1], Pierre Henri[4, 5], Hendrik Ranocha[6], Ingo Richter[1], Martin Rubin[7], and Bruce Tsurutani[8]

[1]Institut für Geophysik und extraterrestrische Physik, Technische Universität Braunschweig, Mendelssohnstr. 3, 38106 Braunschweig, Germany
[2]Max-Planck-Institut fur Sonnensystemforschung, Justus-von-Liebig-Weg 3, D-37077 Göttingen, Germany
[3]ESTEC, European Space Agency, Keplerlaan 1, 2201AZ Noordwijk, The Netherlands
[4]Laboratoire de Physique et Chimie de l'Environnement et de l'Espace (LPC2E), UMR7328 CNRS/Université d'Orléans/CNES, Orléans, France
[5]Laboratoire Lagrange, OCA, UCA, CNRS, Nice, France
[6]Computer Electrical and Mathematical Science and Engineering Division (CEMSE), King Abdullah University of Science and Technology (KAUST), Thuwal 23955-6900, Saudi Arabia
[7]Physikalisches Institut, University of Bern, Sidlerstrasse 5, CH-3012 Bern, Switzerland
[8]Jet Propulsion Laboratory, California Institute of Technology, 4800 Oak Grove Drive, Pasadena, CA 91109, USA

**Correspondence:** Katharina Ostaszewski (k.ostaszewski@tu-bs.de)

**Abstract.** We present a statistical survey of large amplitude, asymmetric plasma, and magnetic field enhancements at comet 67P/Churyumov-Gerasimenko from December 2014 to June 2016. The aim is to provide a general overview of these structures' properties over the mission duration. At comets, nonlinear wave evolution plays an integral part in the development of turbulence and in particular facilitates the transfer of energy and momentum. As the first mission of its kind, the ESA Rosetta mission

was able to study the plasma properties of the inner coma for a prolonged time and during different stages of activity. This enables us to study the temporal evolution of steepened waves and their characteristics. In total, we identified ∼70000 events in the magnetic field data by means of machine learning. We observe that the occurrence of wave events is linked to the activity of the comet, where events are primarily observed at high outgassing rates. No clear indications of a relationship between the occurrence rate and solar wind conditions were found. The waves are found to propagate predominantly perpendicular to the

background magnetic field, which indicates their compressive nature. Characteristics like amplitude, skewness, and width of the waves were extracted by fitting a skew normal distribution to the magnetic field magnitude of individual events. With increasing massloading the average amplitude of steepened waves decreases while the skewness increases. Using a modified 1D MHD model it was possible to show that such solitary structures can be described by the combination of nonlinear, dispersive, and dissipative effects. By combining the model with observations of amplitude, width, and skewness we obtain an estimate of

the effective plasma viscosity in the comet-solar wind interaction region. At 67P/Churyumov-Gerasimenko steepened waves are of particular importance as they dominate the innermost interaction region for intermediate to high activity.





# 1 Introduction

The evolution of nonlinear waves and their influence in the development of weak and strong turbulence is one of the most fun-
damental processes in astrophysical plasmas. Fast and slow magnetosonic waves are known to develop into shocks or solitary
waves. Because of their nonlinear character, these waves interact with the plasma and other waves through a ponderomotive
force, actively altering the ambient plasma in the process. As a result, this may lead to the evolution of inhomogeneities, plasma
heating and the development of turbulence. Hence, they are an integral part of our understanding of the interaction between a
comet and the solar wind. Such waves can be observed predominantly within planetary foreshocks and the cometary interaction
region. In such regions, conditions for the steepening of compressive modes are exceptionally favourable. At Earth, the fore-
shock region has been crossed multiple times by various spacecrafts, which facilitates extensive statistical studies of nonlinear
waves (Tsurutani and Rodriguez, 1981; Schwartz et al., 1992; Giacalone et al., 1993; Mann et al., 1994; Stasiewicz et al.,
2003; Behlke et al., 2004; Lucek et al., 2004). Similar, nonlinear waves were recently studied in great detail at Mars, where the
Maven mission provided high-resolution measurements of magnetic fields and particle properties (Fowler et al., 2018; Shan
et al., 2020). In contrast to planetary missions, cometary missions previous to Rosetta were exclusively fast flybys or impacts,
which only provided limited information on the properties of nonlinear waves for a fixed cometary activity level. First observa-
tions of nonlinear waves at comets were obtained by the International Cometary Explorer (ICE) at 21P/Giacobini-Zinner (Smith
et al., 1986). Tsurutani and Smith (1986) report indications of high-intensity turbulence, which is mainly characterized by the
presence of large-amplitude compressional waves. In the region far upstream of the bow shock predominantly long-period
elliptically polarized waves are present. With decreasing distance to the comet Tsurutani et al. (1987) report steepening of fast
mode waves accompanied by high-frequency wave packets at the leading edge of the wave. In contrast to 21P/Giacobini-Zinner,
where steepening occurred at the leading edge, steepening occurred at the trailing edge in the case of 26P/Grigg-Skjellerup
(Neubauer et al., 1993; Tsurutani et al., 1995). A comprehensive review of low-frequency waves at comets is given by e.g.
Glassmeier et al. (1997).

Unlike previous missions, the groundbreaking ESA Rosetta mission (Glassmeier et al., 2007a; Taylor et al., 2017; Glass-
meier, 2017) was the first of its kind to orbit a comet and study its plasma environment over a prolonged time of about two
years. This provides unprecedented possibilities to observe the evolution of nonlinear waves with special emphasis on the
changing cometary activity level. Due to its operational design, Rosetta was primarily located in the innermost interaction
region and, hence, never able to observe a bow shock crossing. Consequently, the existence of nonlinear waves near the bow
shock region at 67P Churyumov-Gerasimenko (67P/CG) is unconfirmed. Nevertheless, nonlinear phase-steepened waves were
observed in the inner coma of 67P/CG, which is unique compared to previous missions to 1P/Halley, 21P/Giacobini-Zinner
and 27P/Grigg-Skjellerup. Figure 1 shows an exemplary interval of magnetic field data with multiple wave events to showcase
the variability in amplitude and width. It depicts the magnetic field magnitude (top panel) and the magnetic field components
(bottom panel) for a 30 minute time interval, measured on 16 July 2015. During this time interval the spacecraft was located at a
distance of 1.7 au away from the Sun, which is approximately one months before perihelion (August). The outgassing rate was
already high enough to facilitate the development of a diamagnetic cavity (Goetz et al., 2016a, b) and, with a high probability,





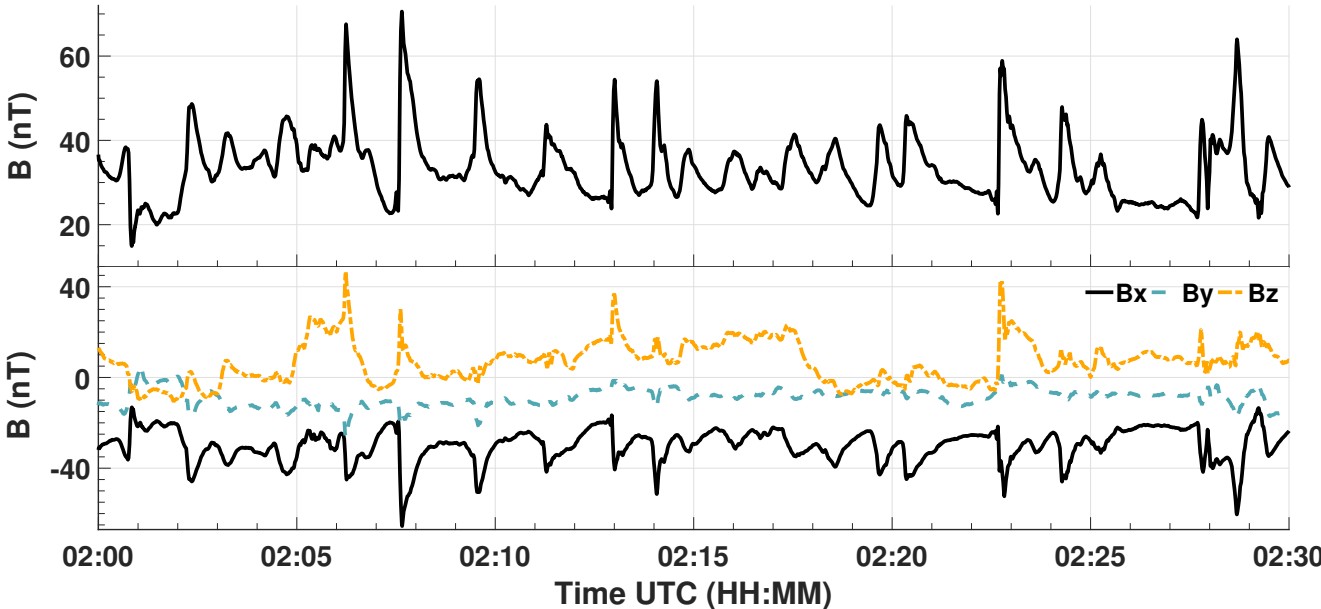

**Figure 1.** Magnetic field magnitude (top panel) and magnetic field components (bottom panel) with multiple occurrences of steepened waves on 16 July 2015.

a bow shock (Koenders et al., 2013, 2015). The striking features of Fig. 1 are the asymmetric, large amplitude enhancements in the magnetic field magnitude. With a background magnetic field strength of around $B_0 = 30$ nT, compression ratios $\delta B / B_0$ range between 1.3 and 2.3. While properties like amplitude, width and strength of asymmetry can change significantly from event to event, they are still strikingly similar in respect to their general shape. Comparable structures have been observed in the electron density (Engelhardt et al., 2018; Hajra et al., 2018b) and ion energy (Stenberg Wieser et al., 2017). Due to their highly asymmetric shape we will refer to them as steepened wave events in the following.

The paper is organized as follows. In Sect. 2 we provide details about instrumentation relevant for this study and the method used to select intervals of interest. Subsequently, the in situ observations of steepened wave events are described and used to characterized the general properties of said wave events in Sect. 3 to Sect. 6. In Sect. 7 we use a modified 1D-MHD model to describe the observed waves as an interplay between nonlinear wave steepening and diffusive effects.

## 2 Instrumentation and event selection

To probe the ambient plasma the Rosetta spacecraft was equipped with a set of five instruments, the Rosetta Plasma Consortium (Carr et al., 2007, RPC), designed to monitor particle properties and electromagnetic fields at 67P/CG. The primary focus of this study is put on the analysis of nonlinear wave signatures in the magnetic field, which were observed by RPC-MAG. The latter consists of two tri-axial fluxgate magnetometer mounted 15 cm apart from each other on a 1.5 m long boom (Glassmeier



et al., 2007b). For the following analysis only the outboard magnetometer data were used. However, the difference between the measured signal on the outboard and inboard magnetometer was used for quality assessment of the magnetic field measurements. If the difference exceeded 5 nT the time interval was excluded from the analysis. The magnetometer can either be
operated in burst mode (20 Hz) or in normal mode (1 Hz). Because of the transient nature of the steepened waves we have exclusively used data sampled with a frequency of 20 Hz. Intervals, where only 1 Hz data were available, were excluded from the analysis. Additionally to the magnetic field, we use electron density data from the RPC Mutual Impedance Probe (Trotignon et al., 2007, RPC-MIP) and neutral gas densities obtained from the Rosetta Orbiter Spectrometer for Ion and Neutral Analysis (Balsiger et al., 2007, ROSINA) comet pressure sensor (ROSINA-COPS) at suitable times. For our analysis we have processed
measurements made between 1 December 2014 to 31 June 2016. In the time periods before and after our interval of interest cometary activity was low, resulting in mostly undisturbed solar wind being observed at Rosetta.

In order to study the occurrence and properties of steepened waves, intervals of interest have to be identified in a first step. The Rosetta magnetic field data (Glassmeier et al., 2019a, b, c, d, e, f, g) used for this study are made available through the PSA archive of ESA (Besse et al., 2018) and the PDS archive of NASA. Intervals of interest can be distinguished by the characteristic
shape, in particular by the distinctly pronounced asymmetry, of the steepened waves in the magnetic field magnitude. Due to the comparatively long mission duration and the resulting large data set, manual identification was found to be unfeasible. Hence, an automated approach using machine learning techniques was used instead (Ostaszewski et al., 2020). Due to their distinct shape and the comparatively large number of wave event occurrences it was possible to train a neural network to detect possible candidates for wave events with a high precision around 80%. Nevertheless, false detections still occurred and were
removed by means of fitting according to the following sections. A list of events is given in the appendix.

To exclude effects introduced by rotation of the spacecraft or comet, the following analysis is performed on data in the Cometary Solar Equatorial (CSEQ) coordinate system. The center of this reference frame is the center of the mass of the comet, the positive x-axis points towards the Sun, the positive z-axis is the component of the Sun's north pole of date orthogonal to the positive x-axis and the y-axis completes the right-handed system. This reference frame is used for all following analysis,
unless otherwise indicated. For all reference system conversions the NASA NAIF SPICE (Acton, 1996) system is used.

## 3   Observations of nonlinear waves

Figure 2 displays the number of detected steepened wave events per week, between 1 December 2014, and 31 June 2016. In the bottom panel, the cometocentric distance and the water outgassing rate, obtained from the Haser model (Haser, 1957) and local neutral gas density measurements from ROSINA-COPS, are shown. In this model a constant neutral gas velocity of 800
$ms^{-1}$ was used. The number of observations is highly variable and can change significantly (factor $\sim$2) on time scales of days. In order to visualize the underlying trend, the observations were organized in week-long bins. From 14 December 2014 to 1 April 2015 and from 1 April 2016 until end of mission RPC-MAG switched multiple times between a sampling rate of 1 Hz (normal mode) and 20 Hz (burst mode). This reduces the number of observations in these weakly active months. This bias was accounted for by correcting the amount of observed waves per week by the fraction of data available during said week. Note



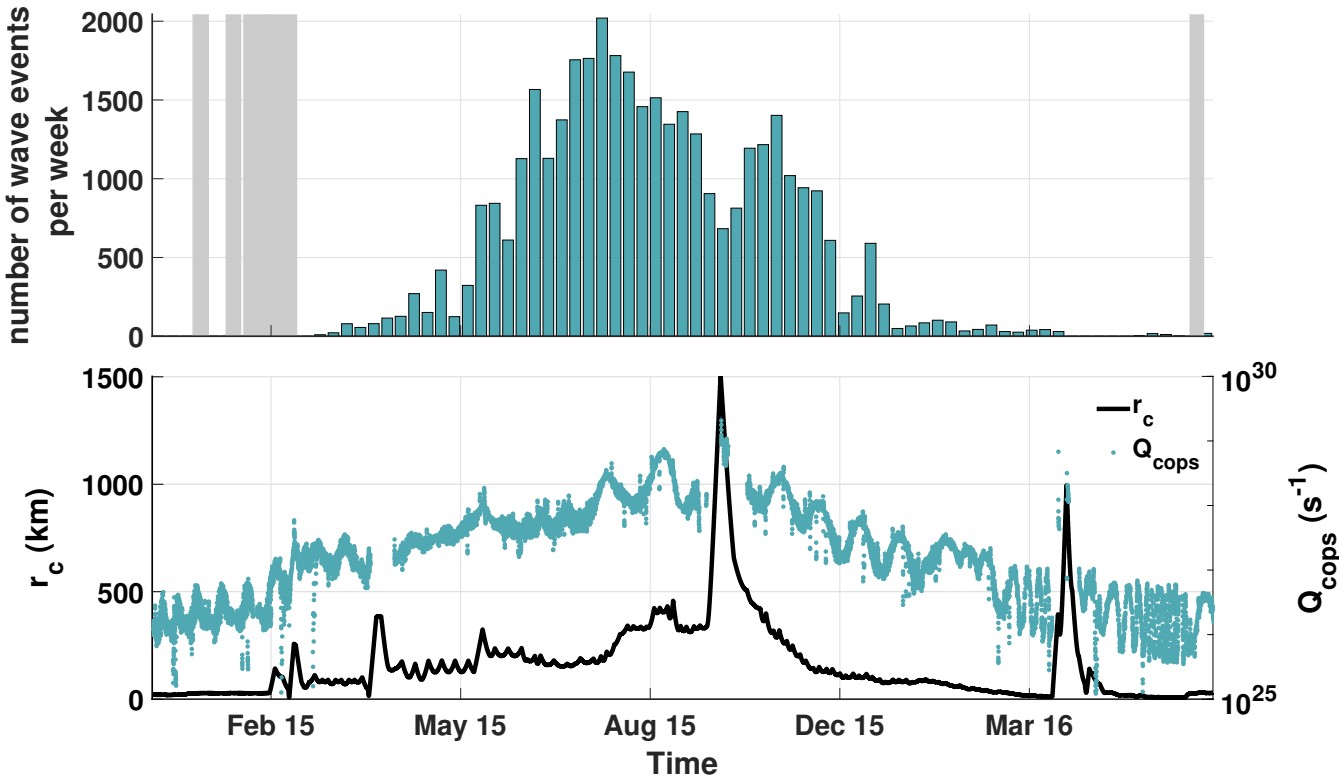

**Figure 2.** The top panel depicts the number of observed steepened waves peer week over the course of the mission. The gray areas mark the periods in which no magnetic field measurements were available. In the bottom panel the water outgassing rate (Q) and the cometocentric distance ($r_c$) are shown. The outgassing rate was computed using the Haser model (Haser, 1957) and local ROSINA-COPS measurements. A constant neutral gas velocity of $800 \ \mathrm{ms^{-1}}$ was assumed. The neutral gas density measurements are noncontinuous with occasional data gaps.

that during late January 2015, early February 2015 and late June 2016 over the course of two weeks no burst mode data were available, resulting in a data gap. The respective time intervals are highlighted in grey.

Shortly after the comet rendezvous on 6 August 2014 low frequency, large amplitude waves were found to dominate the magnetic field observations (Richter et al., 2015). Over the progression of the Rosetta mission these „singing comet" waves (Richter et al., 2015; Richter et al., 2016; Heinisch et al., 2017; Goetz et al., 2020) slowly give way to a more turbulent

interaction region with increasing outgassing rate. During these early month only a few (< 40) isolated instances of steepened wave events were observed. Since the detection method was evaluated for different cometary activity levels, we can exclude that the low number of identified events is due to a bias in the detection method (Ostaszewski et al., 2020). The period from February 2015 to April 2015 marks a transition region in which the „singing comet" signature is not detectable anymore, and first occurrences of cavity crossings (Goetz et al., 2016a) and steepened wave events were observed, which become increasingly





prevalent as cometary activity increased. It is still uncertain if the „singing comet" waves cease to exist or if they are obscured by the increasingly turbulent field closer to perihelion. The exact nature of this transition region and the processes governing it require a more in-depth analysis, which is out of the scope of this paper. Around the time of May 2015, a sudden steep increase in the number of observations above 500 per week is visible. In the month before and after, an average of 100 waves per week is detected with occasional peaks up to 400. From May 2015 until December 2015 the number of wave events, in general, follows

the mean outgassing rate, where a high activity implies a large number of observed waves. Especially on the inbound leg toward the Sun, one can observe how the number of waves steadily increases until it reaches a maximum at the beginning of August 2015, shortly before perihelion. Even though the water production rate further increases after perihelion passage, the number of observations start to decline afterward. Simultaneously with the observed decline, the cometocentric distance increases. This behavior is particularly evident during the dayside excursion in September/October 2015 (Edberg et al., 2016). For this

time interval, a significant decrease of observed waves is visible, while the cometary water outgassing rate further increases. During the dayside excursion, the number of observations declines by half compared to adjacent time intervals. After reaching the furthest distance of 1500 km, Rosetta starts to approach the nucleus again, reaching a distance of 150 km in December 2015. During this time the number of observations increased from around 800 per week to above 2000 per week, even though the water outgassing rate decreased from approximately $10^{29}$ s$^{-1}$ to approximately $2 \cdot 10^{28}$ s$^{-1}$. During the excursion to the

nightside of the comet in March 2016, the number of observed waves was already too low to validate the observed distance dependency during the dayside excursion. It is worth noting that while a general trend in the number of observations is evident, the variations from week to week are still quite large. On occasion, the number of observations can even double compared to neighboring time intervals, while no corresponding signatures are visible neither in cometocentric distance nor in outgassing rate. However, the Haser model (Haser, 1957) assumes a spherically symmetric coma and therefore neglects variations due to

zones of varying activity on the nucleus (Lai et al., 2019).

A better measure for the steepened wave occurrence rate is the solar wind mass-loading. It is the fundamental process governing the solar wind-comet interaction and depends on the cometary activity as well as on the cometrocentric distance(Biermann et al., 1967; Behar et al., 2016; Nilsson et al., 2017; Glassmeier, 2017). The strength of the local mass loading at Rosetta is given by the source term

$$M = m_i \nu_{io} n_n, \tag{1}$$

where $m_i$ is the ion mass, $\nu_{io}$ the ionization frequency and $n_n$ the neutral gas density measured by ROSINA-COPS. Since we use locally measured neutral gas density, variations due to cometary rotation and asymmetric outgassing are taken into account (Hansen et al., 2016; Lai et al., 2019). At 67P/CG the dominant ionisation processes are photoionisation, electron impact ionisation and charge exchange. The importance of the individual processes changes with heliocentric distance and

location in the interaction region. For a strongly active comet and close to the nucleus ($r_c < 1000$ km) the dominant process is photoionisation (Simon Wedlund et al., 2017). The latter varies with distance to the Sun and is given by

$$\nu_{ph} = \frac{\nu_{ph,0}}{r_h^2}, \tag{2}$$



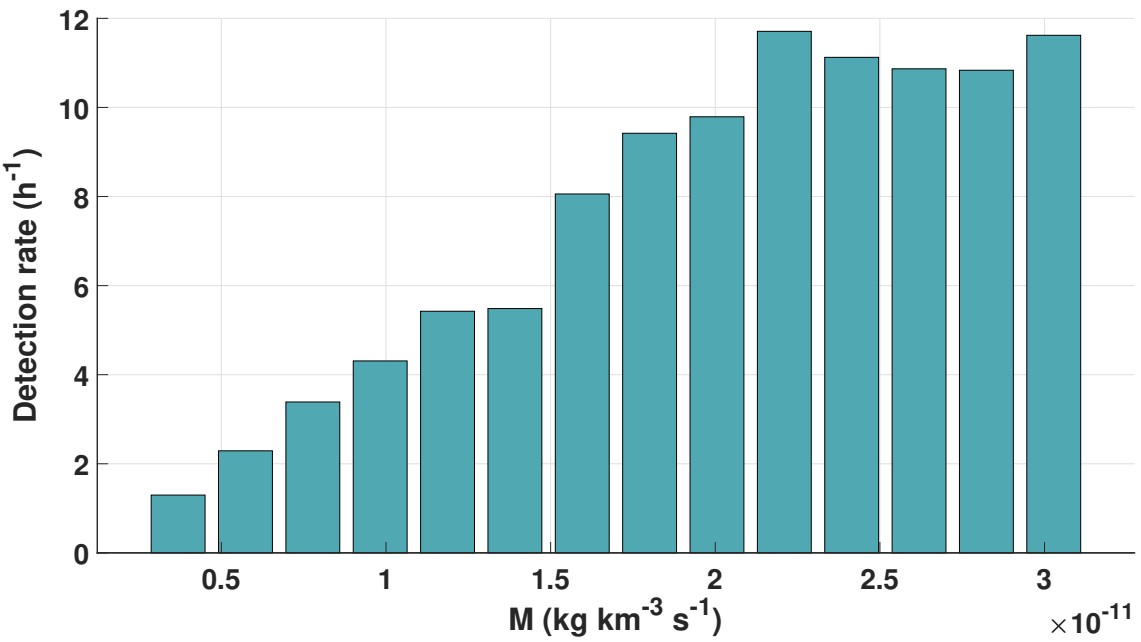

**Figure 3.** Number observed steepened wave events per hour as a function of mass-loading.

where $\nu_{ph,0} = 1.0 \times 10^{-6}$ s$^{-1}$ (Hansen et al., 2007) is the photoionisation frequency at 1 au and $r_h$ is the heliocentric distance in au. Figure 3 shows the number of observed steepened wave events per hour as a function of the mass source term (Eq. (1)).

The rates were obtained by dividing the total number of events observed for a certain mass-loading by the number of hours RPC-MAG operated during the respective local mass-loading conditions. The number of observed events increases linearly with the mass-loading strength up to a certain point, after which the detection rate stagnates at around 11 events per hour.

In general, the variations in the number of observations can be sufficiently explained by changes in the neutral outgassing rate and cometocentric distance (Fig. 2 and Fig. 3). However, in some instances the number of observed wave events abruptly

declines over the course of hours without any noticeable variations in either the neutral gas density or the cometocentric distance. Figure 4 shows the time between two wave event observations between May and September 2015. The time between observations is extremely variable with values between 4 minutes and 78 minutes. Striking are the occasional sharp increases in the time between observations, e.g. on 30 August 2015 the time increased from around 7 minutes to 55 minutes within one day. For these intervals the decrease in detected wave events was manually verified to exclude a bias in the automated

detection procedure. Goetz et al. (2017) and Timar et al. (2019) noted that apart from the neutral gas density, solar wind conditions have a significant influence on the magnetic field at the comet. The dashed line in Fig. 4 shows the solar wind dynamic pressure extrapolated to Rosetta's location with the Tao Model (Tao et al., 2005). In some cases, these sharp increases coincide with increases in the solar wind dynamic pressure. However, most of the time, no correlation between the pressure and time between observations is visible. For the two particular cases marked by arrows, the magnetic field magnitude over a one-



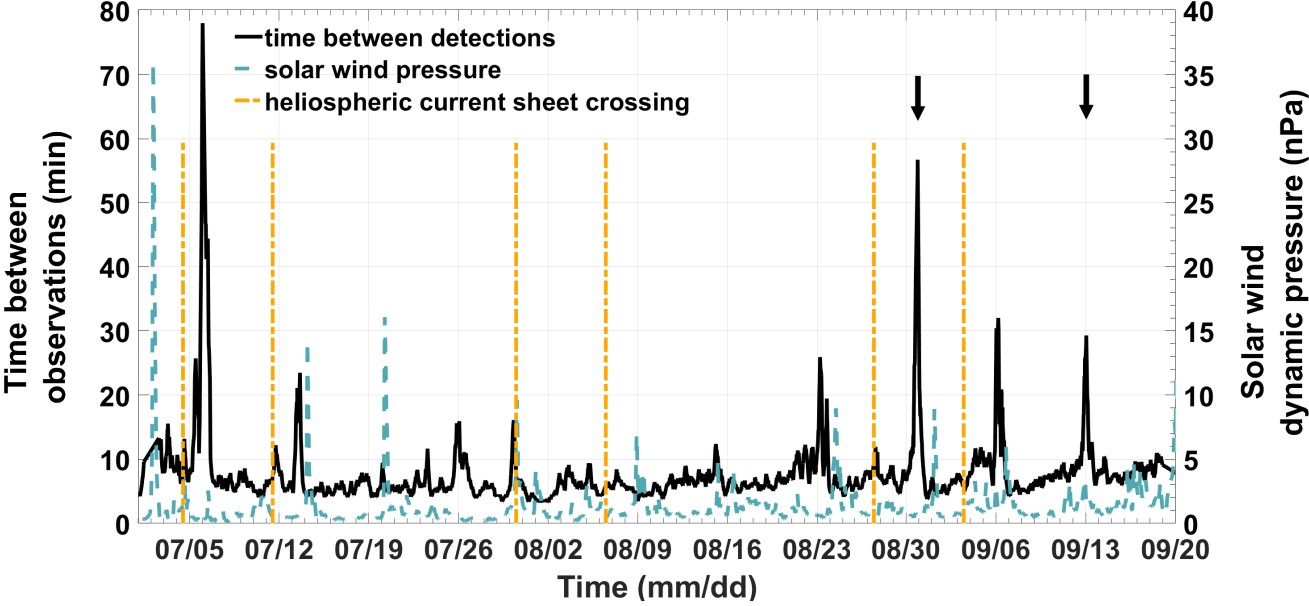

**Figure 4.** Averaged time between detections (solid line) and the solar wind dynamic pressure (dashed line) from 1 July 2015 until 20 September 2015. The solar wind pressure was extrapolated from near-Earth to 67P/CG using the model by Tao et al. (2005). The vertical lines denote the dates of HCS crossings. Occasional sharp increases in the time between detections are visible. Examples of the magnetic field magnitude for the occurrences marked by the arrows are shown in Fig. 5.

hour interval is illustrated in Fig. 5. On 14 September 2015 the magnetic field is dominated by long-period large-amplitude steepened wave events. Shortly before, on 13 September 2015, the interaction region is completely different. Fluctuations occur on significantly shorter time scales with smaller amplitude and the mean magnetic field strength increased from $\sim$20 nT to $\sim$40 nT. The direction of the mean magnetic field stays approximately constant with $\mathbf{B} = (30.99, 16.32, -6.57)$ nT on 14 September 2015 and $\mathbf{B} = (24.53, 5.11, -6.37)$ nT on 13 September 2015. A similar situation can be observed for 29 and

30 August 2015. The previously observed steepened waves are replaced by small scale fluctuations, the mean magnetic field increases from $\sim$26 nT to $\sim$55 nT and the mean field direction only changes marginally from $\mathbf{B} = (-1.34, 31.56, -7.55)$ nT to $\mathbf{B} = (-7.54, 54.85, 18.77)$ nT. The transition between regions occurs smoothly over a time span of multiple hours, as indicated by the smooth increase in the temporal distance between observations (Fig. 4). No sharp changes in the magnetic field, which would indicate a crossing of some boundary, are observed during these time intervals. As Rosetta's position in the interaction

region is approximately constant over a time span of one day, the characteristic increase in mean magnetic field strength is presumably caused by changing solar wind conditions. It hints at a compression of the interaction region, similar to the effects of a interplanetary Coronal Mass Ejection (CME) impact (Edberg et al., 2016; Goetz et al., 2019). During the occurrence of these two examples no apparent changes in the neutral gas density, spacecraft position or solar wind dynamic density, which could explain this complete change of nature of the interaction region, are visible. However, one has to keep in mind that



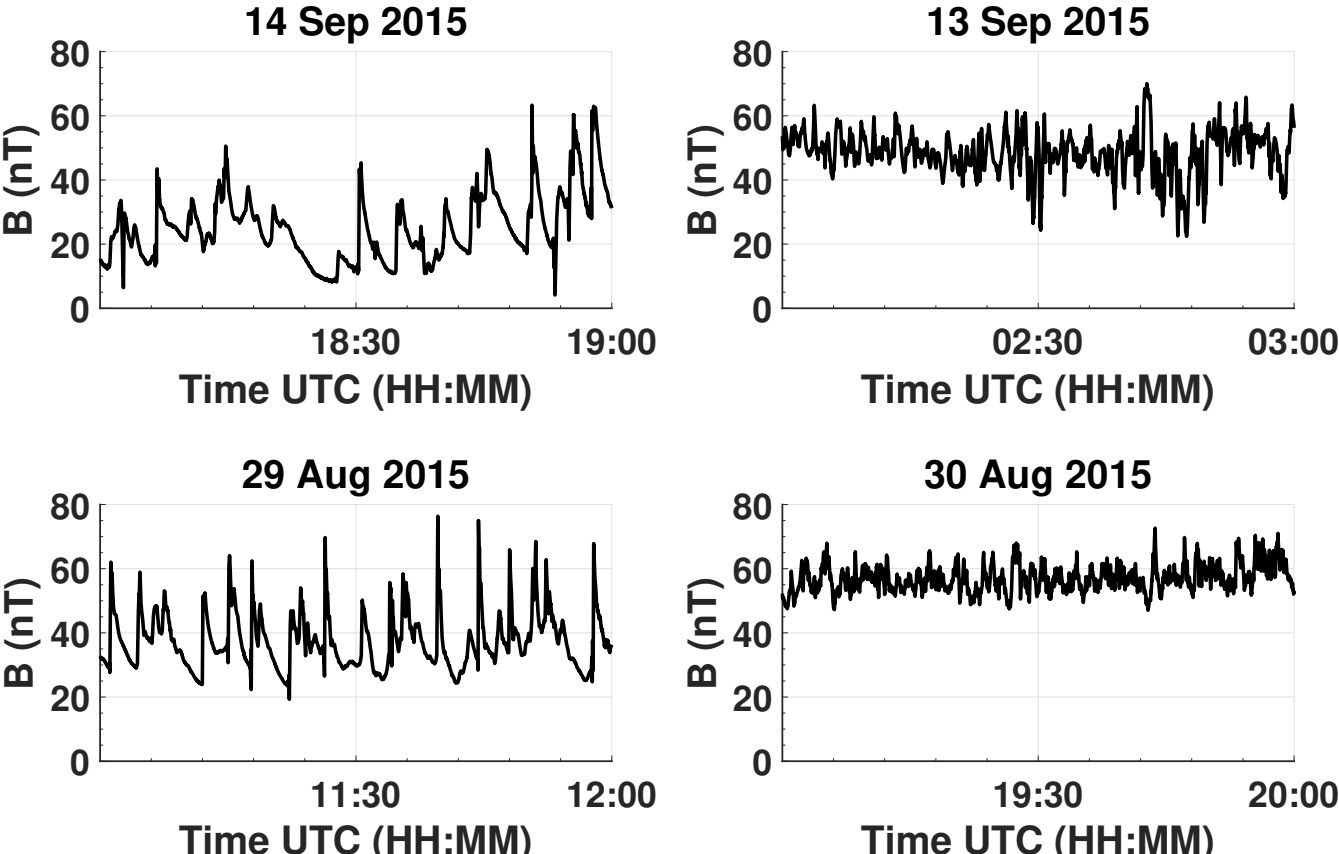

**Figure 5.** On the left typical examples of an interaction region dominated by steepened waves are shown. On the right the interaction region one day later or earlier is shown. Instead of steepened waves, oscillations on smaller scales and with significantly smaller amplitude are visible, which resemble 1P/Halley's interaction region (Glassmeier et al., 1997). The illustrated time intervals correspond to the highlighted peaks in detection time in Fig. 4.

the solar wind observations are not in-situ measurements, but rather measurements obtained near Earth and extrapolated to Rosetta's location with the Tao model (Tao et al., 2005) and may therefore be inaccurate. Another possible explanation are transient solar wind events. Candidates are CMEs, Co-rotating Interaction Regions (CIRs) and Heliospheric Current Sheet (HCS) crossings (Smith et al., 1978). Both CMEs (Edberg et al., 2016; Goetz et al., 2019) and CIRs (Hajra et al., 2018a) are known to compress the cometary interaction region and cause such increases in the magnetic field. Since HCS crossings are

a reversal of the interplanetary magnetic field, it is unclear if they could affect the cometary interaction region in a significant way. However, adjacent to HCSs are very high plasma densities, which are likely able to compress the cometary interaction region in a similar way to CMEs and CIRs (Tsurutani et al., 2016). For the considered time intervals, no CME or CIR events were observed. As a reference the Rosetta science events list was used (Rosetta Team, 2020). A list of HCS crossings is given by the Wilcox Solar Observatory (Svalgaard and Wilcox, 1976; Svalgaard, 2020). The dates of crossings are marked by orange





vertical lines in Fig. 4. A direct correlation between HCS crossings and increases in time between observations is not visible. However, in most cases an increase in the time between two detected wave events occurs within days after a HCS crossing. Overall, to determine the governing processes for these changes in the interaction region a more in-depth analysis of the local solar wind conditions using different models and databases is necessary. Since this is out of the scope of this paper, it is left for further research.

## 4   Types of waves


Based on the shape of the waves in the magnetic field magnitude, two types of wave events can be identified. In Fig. 6 (a), the prototypical steepened wave event is displayed. It is characterized by a sharp increase in magnetic field magnitude on time scales of seconds to minutes, followed by a more gradual decline. The steep leading edge is typically observed first. The second type of event principally resembles the first, with the addition of oscillations at the leading edge. These types of dispersive

effects are evident to varying degrees. In a weakly pronounced case, the dispersive effects cause an undershoot, which can vary between several nT up to 10s of nT, at the foot of the leading edge. In a more developed state, multiple oscillations at the leading edge are present, for which frequency and amplitude visibly decrease with distance from the edge. These dispersive effects resemble the whistler packets observed at 21P/Giacobini-Zinner (Tsurutani et al., 1987). It is worth noting that while the degree of dispersive effects differs significantly, the overall shape of the waves is still remarkably alike. Steepened wave

events with dispersive effects constitute around 40 % of all observations. Hereby, weakly pronounced effects are most frequent, whereas strong effects similar to Fig. 6 (b) are comparably rare. Compared to observations at 21P/Giacobini-Zinner, where the high-frequency whistler packets accompanied nearly every steepened wave, this percentage is remarkably low. Instances in which the oscillations are visible behind the steep edge can also be observed occasionally.

## 5   Steepened wave characteristics

As is evident from Fig. 1, the amplitude, width and asymmetry of the waves varies significantly. In order to derive these quantities in a consistent manner a skew-normal distribution (SND, Eq. 3 - 5), as given by Azzalini (1985) is fitted to the magnetic field magnitude in intervals of interest. The SND was chosen because it is similar in shape to the steepened waves and provides comprehensive definitions of amplitude, skewness and width. Moreover, by using a fit instead of directly computing amplitude and skewness from the magnetic field measurements, the coefficients of determination provide information on how

well a wave event resembles a steepened wave.

    Usually, the skew-normal distribution is described by three parameters, the location $\tilde{x}$, the scale $\delta$ and the shape $\alpha$. In this case, two additional parameters $B_0$ and $B_{amp}$, that account for a background magnetic field and scaling to actual field strength





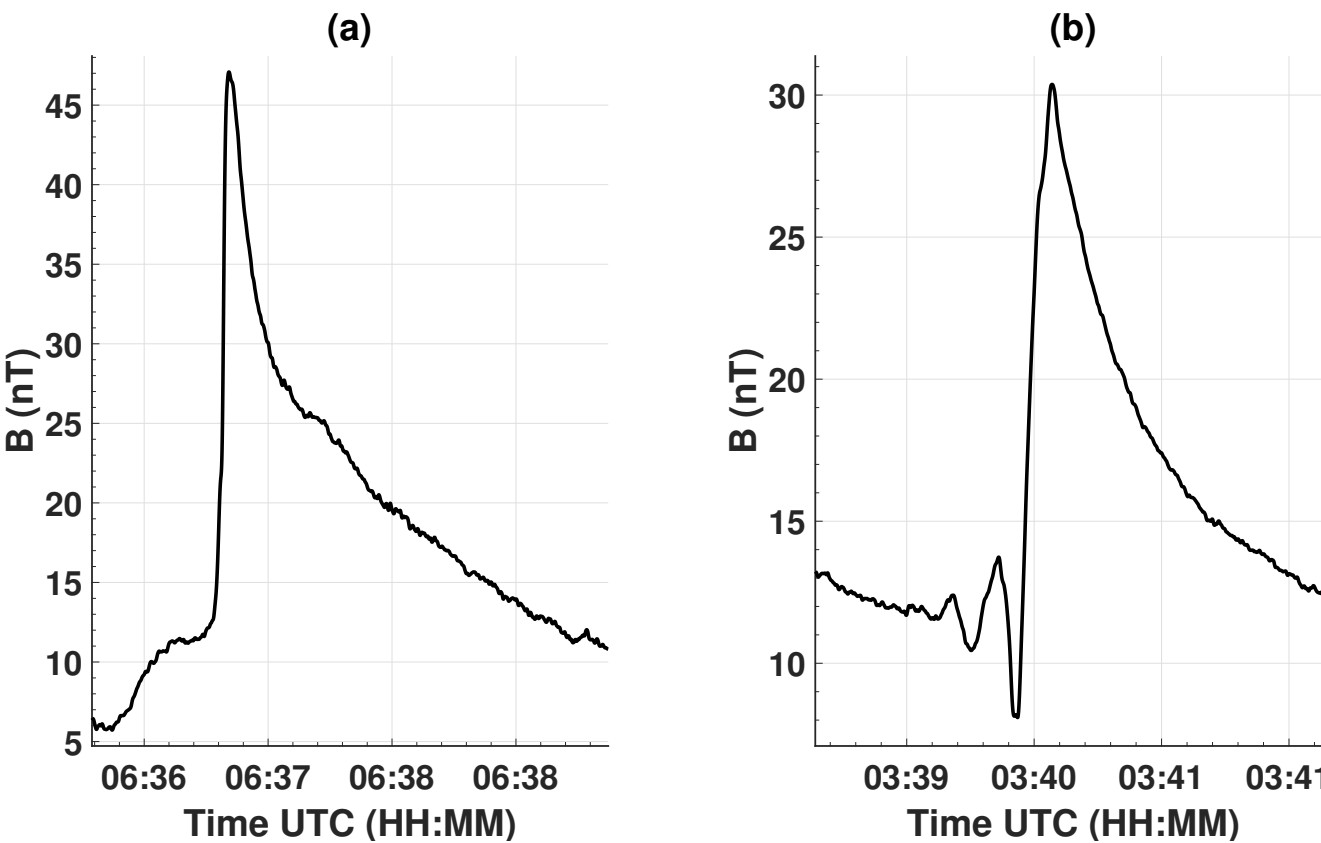

**Figure 6.** Example of a typical steepened wave without dispersive effects on 20 November 2015 (panel a) and with dispersive effects on 3 August 2015 (panel b).

respectively, are added:

$$\Phi(x) = \frac{1}{\sqrt{2\pi}} \exp\left(-\frac{x^2}{2}\right) \tag{3}$$

$$\Theta(x) = \frac{1}{2}\left[1 + \mathrm{erf}\left(\frac{\alpha x}{\sqrt{2}}\right)\right] \tag{4}$$

$$f(x) = \frac{2B_{amp}}{\delta}\Phi\left(\frac{x - \tilde{x}}{\delta}\right)\Theta\left(\alpha\left(\frac{x - \tilde{x}}{\delta}\right)\right) + B_0. \tag{5}$$

To illustrate the validity of the approach we first compute an average shape of the steepened wave events using a superposed epoch analysis (Chree, 1913, SEA) and then fit a skew-normal distribution to the average shape using the Levenberg-Marquardt-Algorithm (Marquardt, 1963). For the averaging process we used unfiltered data to ensure that the asymmetric shape, in particular the steep edge, of the wave is not affected. As amplitude and width of events vary widely, the amplitude was normalized and the intervals of interest resampled to the most common steepened waves width of 35 s. Moreover, we subtracted the mean magnetic field magnitude to account for varying offsets. Normalization of the magnitude is essential since otherwise





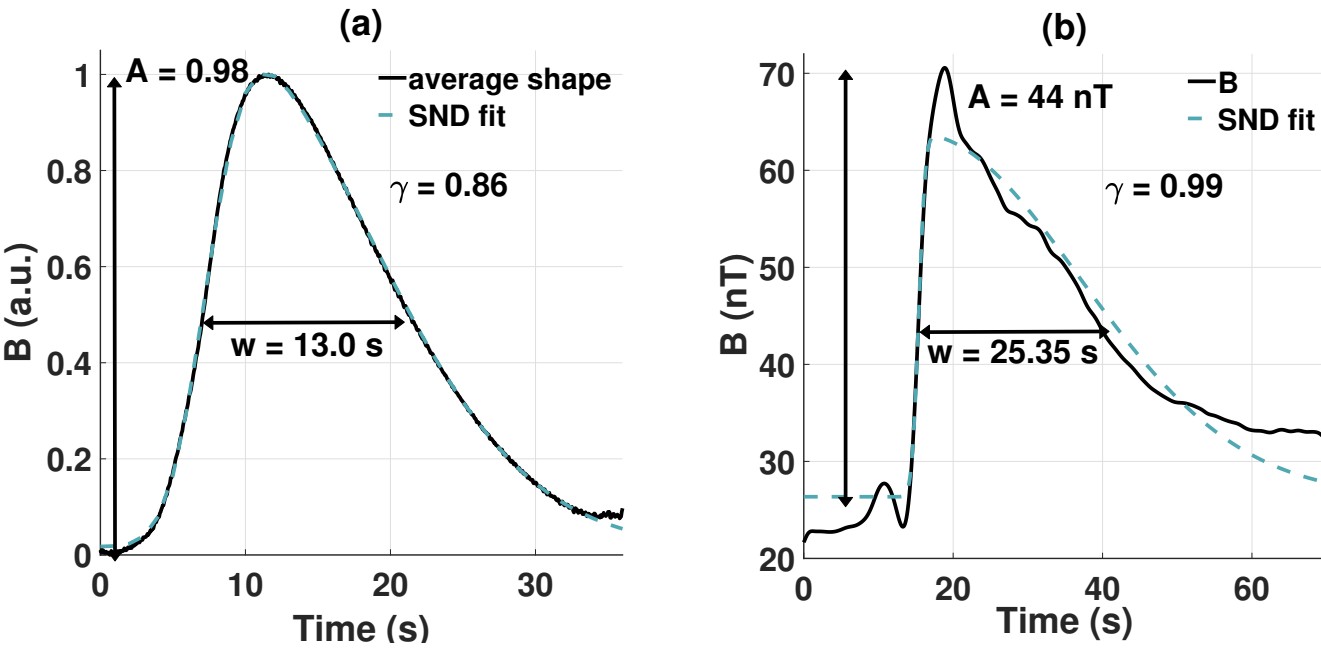

**Figure 7.** Skew normal distribution fitted to the average shape (left) and an exemplary steepened wave (right). Through the skew normal distribution comprehensive definitions of amplitude, skewness and width are obtained.

the average shape will be skewed towards steepened waves with a large amplitude. The signals are then temporally aligned by computing the shift using cross-correlation. Figure 7 (a) shows the average shape of steepened waves obtained by the SEA. As

can be seen, the SND can capture the general characteristics of the steepened waves adequately with an adjusted R-squared value of 0.98. For the following analysis, only fits with an adjusted R-squared value above 0.7 were taken into account. Waves with skewness values below 0.6 were discarded because they are not sufficiently asymmetric to be considered steepened. This reduces the number of events from initially approximately 70000 to approximately 45000.

The skewness $\gamma$ and amplitude $A$ of the steepened wave are obtained by

$$\gamma = \frac{4 - \pi}{2} \left( \frac{\delta \sqrt{2/\pi}}{\sqrt{1 - 2\delta^2/\pi}} \right)^3 \tag{6}$$

$$A = \sqrt{\frac{2}{\pi}} \delta - \frac{\gamma \sqrt{1 - 2\delta^2/\pi}}{2} - \frac{\text{sign}(\alpha)}{2} \exp\left( -\frac{2\pi}{|\alpha|} \right), \tag{7}$$

where $\delta = \alpha/\sqrt{1 + \alpha^2}$ (Azzalini, 1985). Values for the skewness range from -1 for left skewed distributions to 1 for right skewed, where a value of 0 signals a symmetric distribution. As a measure of the width the full width at half maximum (FWHM) is chosen, as it provides a well defined point at which the widths can be compared. While the average shape is

described well by the SND, individual steepened waves can differ slightly in shape compared to the SND. In some cases this can lead to errors in the determination of the amplitude, as shown in Fig. 7 (b). Hence, the amplitude is computed as the difference between the steepened wave maximum and the footpoint in the magnetic field data.



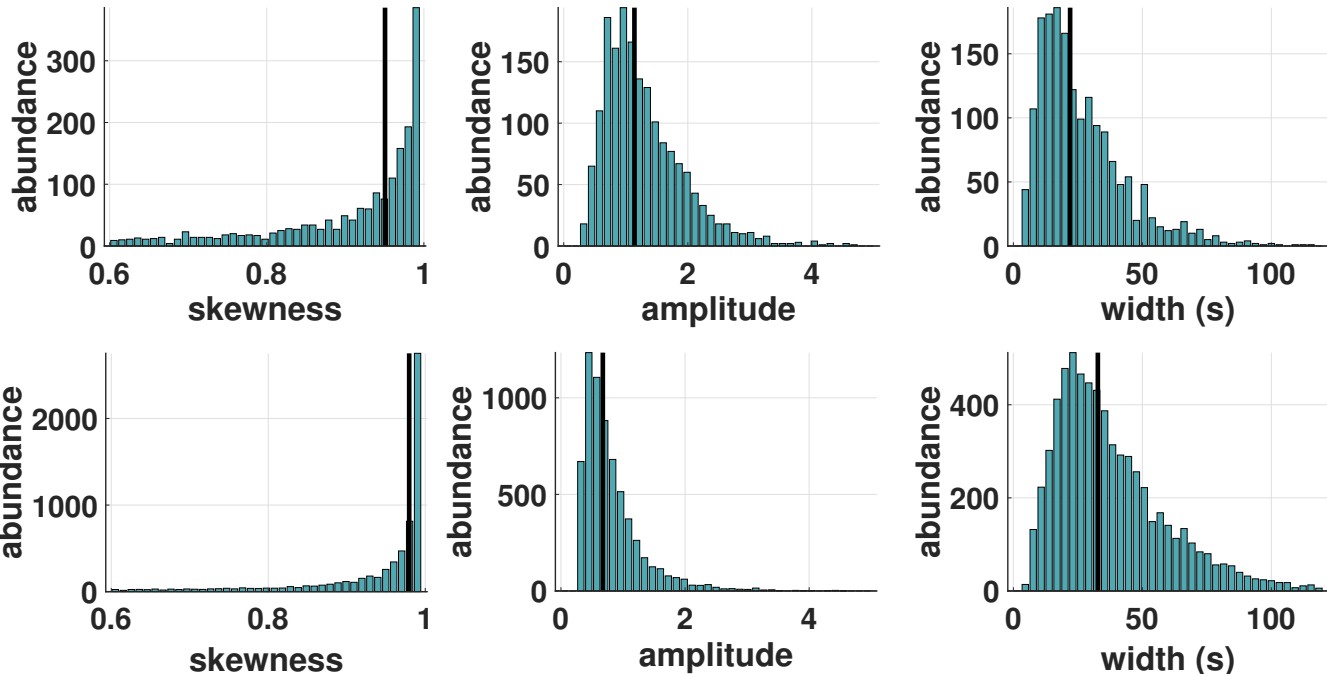

**Figure 8.** Distributions of skewness, amplitude and width for different levels of cometary activity. In the top panel, the properties of steepened waves for low to intermediate activity from 1 March 2015 to 1 June 2015 ($Q \sim 1 \times 10^{27} s^{-1}$). For a strongly active comet from 15 August 2015 to 15 September 2015 ($Q \sim 2 \times 10^{28} s^{-1}$) the properties are shown in the bottom panel. The black line marks the median of the respective distribution.

Figure 8 shows the distributions of amplitude, skewness and width for two different levels of activity. The amplitude was normalized to the magnetic field strength at the footpoint of the wave. The black line marks the median of the respective distribution. The top panel depicts the distribution from 1 March 2015 until 1 June 2015, which corresponds to an intermediately active comet. In the bottom panel, the distribution moments for the duration of one month from 15 August 2015 until 15 September 2015, for a highly active comet are shown. Independent of the activity level, waves with low and high skewness values can always be observed, with a general trend towards higher values. With rising activity, the distribution leans more towards higher skewness values, which can be quantified by the median of the distribution. In Fig. 8, it can be seen that the median rises from 0.93 to 0.98, showing that the number of waves with high skewness increases with cometary activity. The distribution of amplitudes ranges from values around 0.3 to around 4, which shows that the waves are highly nonlinear. However, in contrast to the median of the skewness, which increased with activity, the amplitude median decreases from 1.2 to 0.8. The absolute amplitude median, on the other hand, increases slightly from 18 nT to 22 nT. Lastly, more waves with larger width can be observed at higher activity. Note that the width can only be measured in time since information about the propagation velocity is not available. Therefore, differences in width are only partially features of the steepened waves, but rather changes in the bulk velocity of the propagation medium can also cause these variations.





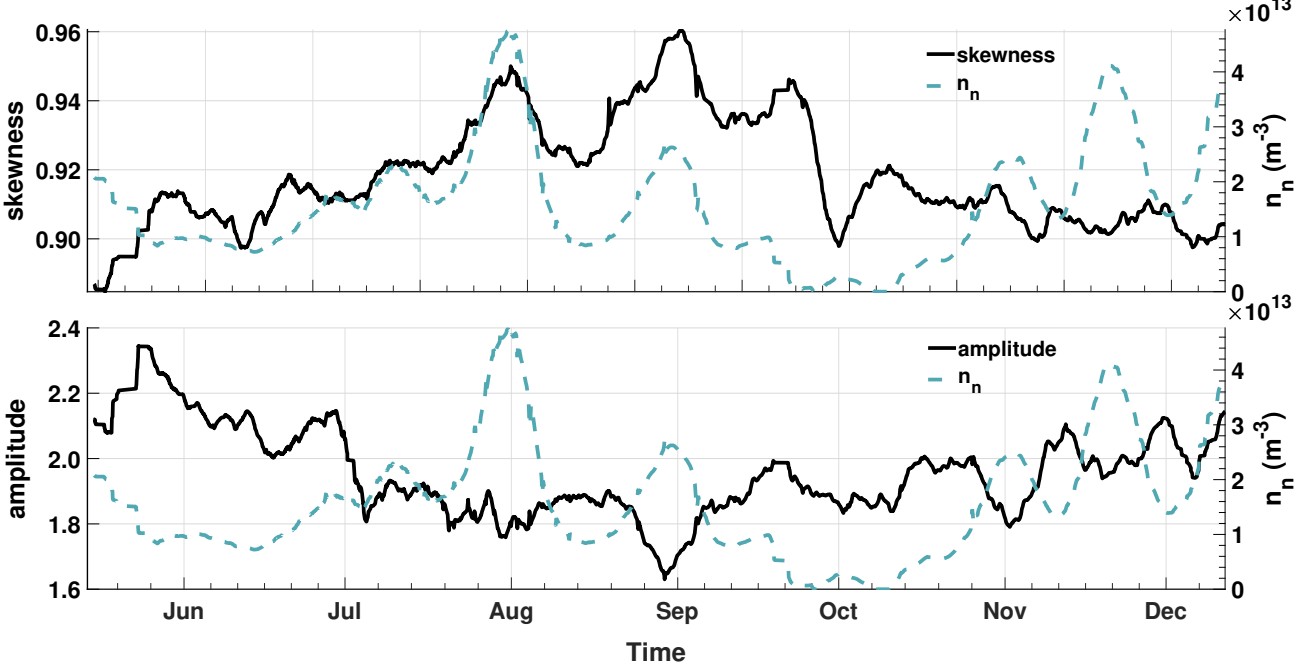

**Figure 9.** Averaged skewness and neutral gas density from 14 May 2015 to 12 December 2015 (top panel). In the bottom panel the averaged amplitude and neutral gas density are shown. While the skewness exhibits a clear correlation with the neutral gas density, especially until October, the amplitude is mostly anti-correlated to the neutral gas density.

Figure 3 and Fig. 8 illustrate that the properties of the steepened waves are governed by variations in the local plasma environment. Hence, the local plasma density and neutral gas density are expected to influence the development of the waves. In the top panel of Fig. 9, the skewness and neutral gas density, averaged over one week, are shown as a function of time. Until
late September 2015, the skewness and neutral gas density are in good agreement with a correlation coefficient of $\rho = 0.76$. From September onwards, the correlation ceases. In contrast, the amplitude shows an apparent anti-correlation with $\rho = -0.61$, which is especially evident around September.

## 6  Wave normal direction

The minimum variance analysis (MVA) by Sonnerup and Cahill (1967) and Sonnerup and Scheible (1998) is a method fre-
quently used to determine the wave propagation direction. To determine the normal direction of the steepened wave, in a first step, a sixth-order Butterworth lowpass filter (Butterworth, 1930) with a cutoff at 500 mHz was applied to the magnetic field observations to exclude high-frequency oscillations. The cutoff frequency was chosen such that the steep leading edge was unaffected by the filter. In order to exclude waves which do not exhibit a distinct pattern only results with a corresponding





eigenvalue ration of $40 > \lambda_{med}/\lambda_{min} > 5$ are selected. The waves are in general well defined with a mean eigenvalue ratio
$\lambda_{med}/\lambda_{min}$ of 13.7.

As a reference point, the angle between minimum variance direction and the local cometary background field is calculated. For such a turbulent interaction region as the inner coma, it is difficult to define what constitutes a background field. Following Goetz et al. (2017) the background field is assumed to be the mean magnetic field, which was obtained by applying a sixth-order Butterworth lowpass filter with a cutoff frequency of 0.1 mHz followed by averaging over a sliding window with a size
of 20 minutes and a displacement of 10 minutes to the magnetic field components. For the cutoff frequency, a value was chosen so that all local disturbances, especially the steepened waves, which are very broad in the frequency space, were removed, while global variations caused by, e.g. diurnal changes remained visible. As the directions of minimum variance, as well as the cometary background field, are susceptible to offsets in the components, the following analysis was only performed for the periods in which diamagnetic cavities were available to adjust the offsets. Therefore, time frames before 20 April 2015, during
the day-side excursion and after 17 February 2016 are excluded.

The angle $\Phi(\boldsymbol{a},\boldsymbol{b})$ between two vectors $\boldsymbol{a}$ and $\boldsymbol{b}$ can be calculated following

$$\Phi(\boldsymbol{a},\boldsymbol{b}) = \arctan\left(\frac{\|\boldsymbol{a} \times \boldsymbol{b}\|}{\boldsymbol{a} \cdot \boldsymbol{b}}\right). \tag{8}$$

Because of the ambiguity of the MVA to the sign, the range of values for the angle $\Phi$ spans over $[0°, 180°]$. As mentioned by, among others, Narita (2017) the minimum variance analysis fails for linearly polarized waves, because the polarization plane
is not uniquely determined. Therefore, all steepened waves with an ellipticity above 0.9 are disregarded for the analysis of the wave propagation direction. In general, the steepened waves are highly elliptically polarized with a mean ellipticity of 0.7. As a consequence of this approach, the number of valid events is significantly reduced to around 15000.

The histogram in Fig. 10 (a) shows the abundance of angles between the minimum variance direction of the steepened waves and the cometary background field. Due to the larger circumference on a sphere for $\Phi \approx 90°$ than for $\Phi \approx 0°$, the number of
observed angles will be biased towards $90°$. To remove this bias, the number of observations $N_i$ for bin $i$ were multiplied by $\sin(\phi_i)$ where $\phi_i$ is the corresponding angle for bin i. As can be seen the minimum variance direction follows a remarkably well defined normal distribution with a mean of $\mu = 89.88°$ and a standard deviation of s $= \sqrt{\sigma} = 21.50°$. Thus, the waves propagate predominantly perpendicular to the cometary background field. However, as Eq. (8) defines an angle in 3D space it is invariant to rotation along the background magnetic field direction. Therefore, the direction of the minimum variance in the
plane perpendicular to the background field vector is uncertain by $180°$. As a second point of reference, the angle between the minimum variance direction and the Sun is shown in Fig. 10 (b). The distribution has a bimodal shape with maxima around $65°$ and $115°$ or $\pm 65°$ due to the MVA ambiguity. This does not contradict the fact that the waves propagate perpendicular to the background magnetic field, since for a strongly outgassing comet the magnetic field drapes around the comet (Goetz et al., 2017; Volwerk et al., 2018). Consequently, the background magnetic field is predominantly oriented in the x-direction
sun- or antisunwards. The asymmetric shape may be introduced by orbital configuration and is therefore not necessarily of scientific interest. Figure 11 shows Rosetta's trajectory from 5 June 2015 to 15 August 2015 and the propagation direction of the steepened waves indicated by black arrows. During this time interval, Rosetta was mainly in a terminator orbit with





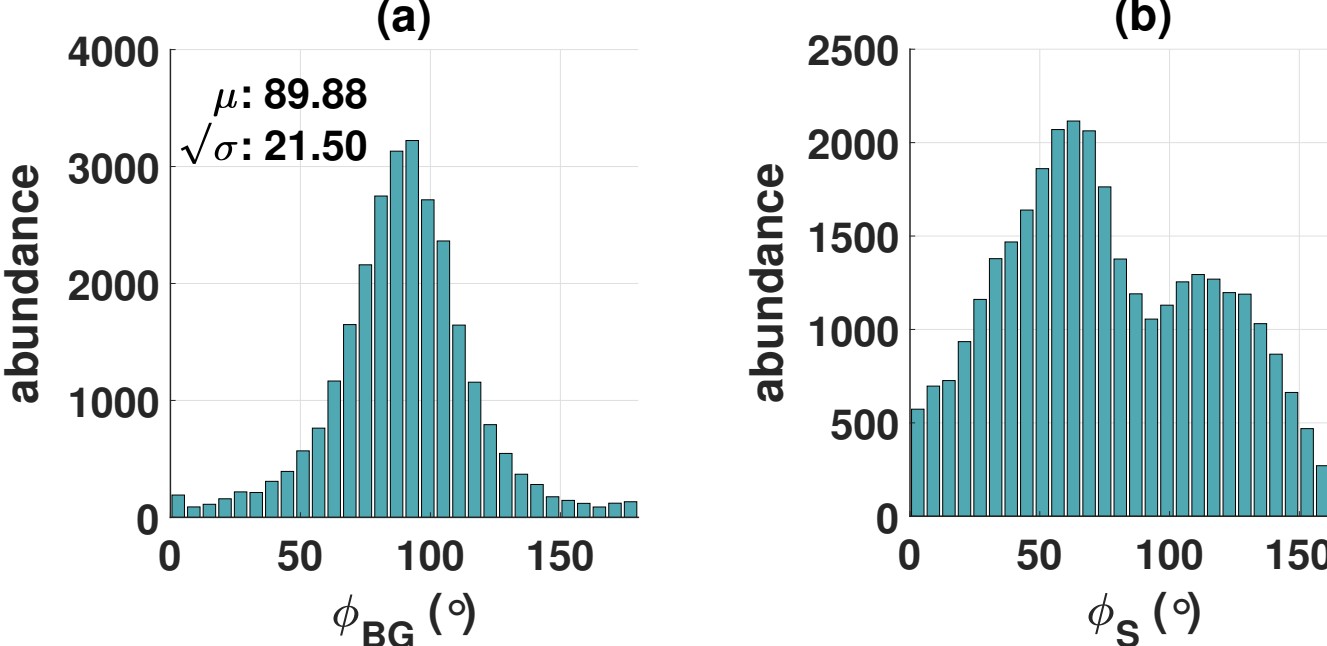

**Figure 10.** Histograms of the angle between the propagation direction and the background magnetic field (a) and the spacecraft-Sun connection line (b). The waves propagate predominantly perpendicular to the background field and at an angle $\pm 65°$ to the Sun.

minimal variation in the x-direction. In the figure Rosetta's trajectory and the propagation direction of the steepened wave events are projected onto the yz-plane of the CSEQ-coordinate system. The minimum variance direction was adjusted so that

every instance has the same orientation, which is arbitrarily chosen to be oriented away from the nucleus. A moving average for a time interval of two hours is then applied to the minimum variance direction. To further increase the visibility, only every 10th vector is plotted. It is clearly visible how the waves change their propagation direction over the course of Rosetta's trajectory, so that, in the Fig 11 , they are oriented approximately away from the comet. The pattern that the minimum variance direction exhibits in Fig. 11 resembles the general motion of cometary ions (< 60 eV) close to the nucleus (Odelstad et al., 2018; Nilsson

et al., 2020). A similar flow pattern was also found for accelerated ions (40 - 80 eV) inside and outside of the diamagnetic cavity (Masunaga et al., 2019). In both cases a significant antisunward motion of the ions toward the tail was reported. Due to the ambiguity of the MVA it is unclear if the waves have a sun- or antisunward motion. In the configuration chosen in Figure 11 the waves exhibit a slight sunward motion. The propagation direction perpendicular to the background magnetic field and compressive nature of these waves (Engelhardt et al., 2018; Hajra et al., 2018b) are clear indicators that they behave like fast

magnetosonic waves.

Main properties of these waves can be summarized as follows:

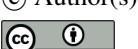



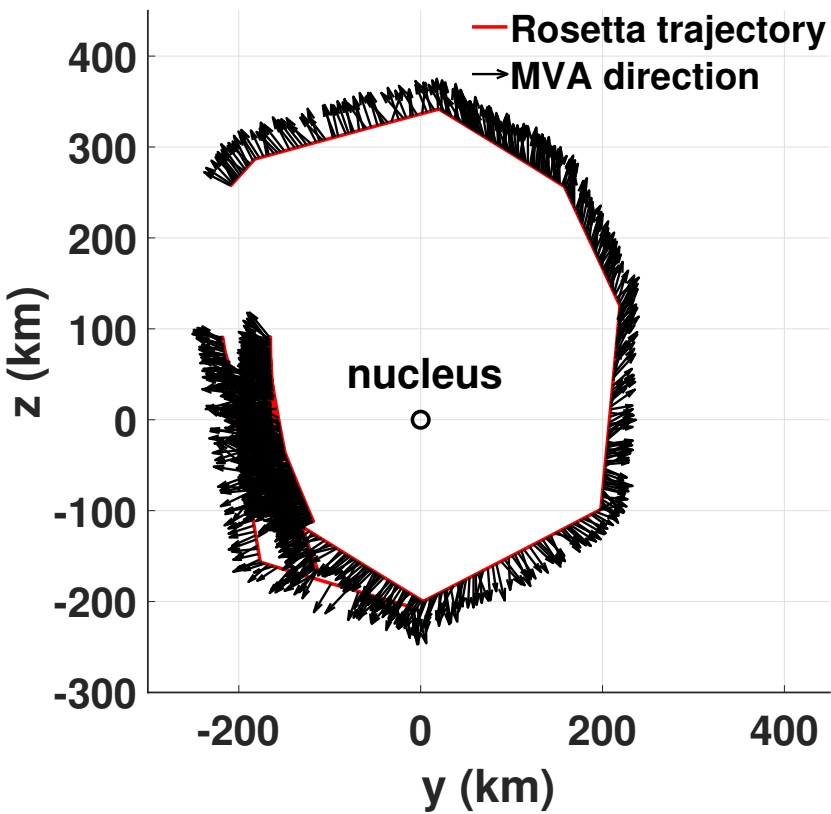

**Figure 11.** Illustration of the propagation angle projected onto the yz-plane of the CSEQ frame for the time interval from 5 June 2015 to 15 August 2015. The red solid line at the base of the black arrows denotes the spacecraft trajectory, while the black arrows show the wave propagation direction. The vectors were adjusted so that all have the same orientation, which was arbitrarily chosen to be pointing away from the nucleus.

1. The number of observed steepened wave events predominantly depends on the mass-loading. Influences of extreme solar wind conditions can be seen in occasional sudden increases of the time between observation of two events.

2. Steepened wave events can be grouped into two categories, those events with dispersive effects and those without. In the former case, high frequency oscillations are visible at the leading edge of the wave.

3. Skewness and amplitude of the steepened waves depend on the neutral gas density, where the former increases with density and the latter decreases.

4. Based on an MVA, these waves propagate perpendicular to the background magnetic field and at an angle of $\pm 65°$ to the Sun. The propagation direction of these waves resembles the flow pattern of cometary ions close to the nucleus. Propagation perpendicular to the background magnetic field is typical for fast magnetosonic waves.







# 7 Theoretical modelling

To understand the basic properties of the wave events described above we use a modified 1D-MHD model. Since the observed non-linear waves predominantly propagate perpendicular to the background magnetic field the 1D assumption is justified. As the inner coma is characterized by a high neutral gas density in comparison to the plasma density and the wave processes

are sufficiently slow, it is essential to take neutral gas effects into account as well. This leads to additional damping effects based on ion-neutral and electron-neutral collisions. In general, the damping rate is a function of the wave frequency and will therefore affect the skewness of the wave. Thus, we can obtain an estimate of the local effective wave damping rate by using the observations of skewness and amplitude and by modeling the wave evolution using a modified 1D-MHD model.

During the high activity phase, the plasma in the innermost interaction region is predominantly of cometary origin. The

ion composition close to the nucleus changes with heliocentric and cometocentric distance. However, the three dominant species $H_3O^+$, $NH_4^+$ and $H_2O^+$ all have a mass to charge ratio of 18 - 19 u/e (Heritier et al., 2017). Hence, we model the charged plasma component using a single fluid with a mass to charge ratio of 19 u/e. The behavior and damping rates of waves in a partially ionized plasma are heavily influenced by the complex interaction between ions and neutrals (Zaqarashvili et al., 2011; Soler et al., 2013; Vranjes, 2014; Martínez-Gómez et al., 2018). Among the main dissipation mechanisms are

resistive dissipation, viscous dissipation and ion-neutral friction. These mechanisms depend on the properties of the ambient plasma, in particular on the ion-neutral or electron-neutral collision frequencies. To correctly model this behavior a multi-fluid model describing the interaction between the charged and neutral fluid is necessary. However, due to temporarily and spatially limited plasma measurements (in particular ion velocities) the wave evolution, especially the ion-neutral interaction, cannot be sufficiently resolved. Hence, with only limited information available, such a high level of theoretical detail is impractical.

Moreover, the dynamics of the neutral fluid beyond the ion-neutral and electron-neutral interaction is not of interest for this study. Thus, to reduce the complexity of the model we parametrize the wave damping using an effective viscosity and resistivity. Then the additional dissipation induced by the ion-neutral and electron-neutral interaction can be approximated without going into too much detail about the underlying physical processes. We consider two different mechanisms, resistive and viscous damping, since they depend on different plasma parameters. By comparing the values obtained through simulations with

suitable reference values, constituent processes influencing the wave damping can be identified. Assuming $\boldsymbol{B} = (0, 0, B)$, $\boldsymbol{u} = (u, 0, 0)$ and $\nabla \equiv \partial/\partial_x$ the 1D fluid equations with resistive and viscous contributions are (Warburton and Karniadakis,





1999):

$$\frac{\partial \rho'}{\partial t'} = -\frac{\partial}{\partial x'}\left(\rho' u'\right) \tag{9}$$

$$\frac{\partial \left(\rho' u'\right)}{\partial t'} = -\frac{\partial}{\partial x'}\left(\rho' u'^2 + p' + \frac{B'^2}{2} - \frac{\nu'}{3}\frac{\partial u'}{\partial x'}\right) \tag{10}$$

$$\frac{\partial E'}{\partial t'} = -\frac{\partial}{\partial x'}\left(u'\left(E' + p' + \frac{B'^2}{2}\right) - \frac{\nu'}{6}\frac{\partial u'^2}{\partial x'} - \frac{\eta'}{2}\frac{\partial B'^2}{\partial x'}\right) \tag{11}$$

$$\frac{\partial B'}{\partial t'} = -\frac{\partial}{\partial x'}\left(u' B' - \eta'\frac{\partial B'}{\partial x'}\right). \tag{12}$$

where the total energy density is given by

$$E' = E'_{int} + \frac{1}{2}\rho' u'^2 + \frac{B'^2}{2}. \tag{13}$$

Herein, $B'$ is the magnetic field, $u'$ the plasma bulk velocity, $\rho'$ the mass density, $p'$ the pressure, $E'_{int}$ the internal energy, $\eta'$ the resistivity and $\nu'$ the kinematic viscosity. The prime denotes normalized quantities, where the mass density is normalized by the equilibrium mass density $\rho_0$, the bulk velocity by the Alfvén speed $v_A$, the magnetic field by $B_0$, the time by the ion gyroperiod $\Omega_i^{-1}$, space by the ion skin depth $v_A * \Omega_i$, the resistivity $\eta$ by $\frac{\Omega_i}{\mu_0 v_A^2}$ and the kinematic viscosity $\nu$ by $\frac{\Omega_i}{\rho_0 v_A^2}$.

In this model, the wave propagates perpendicular to the background field. Therefore, only the fast mode is described. Moreover, the Hall-Term in the induction equation vanishes. Consequently, dispersive effects as seen in Fig. 6 are not modeled. However, this set up has the advantage that the numerical solution for the magnetic field is inherently divergence-free so that no additional divergence cleaning steps have to be taken (Ranocha et al., 2020). Due to the nonlinear terms an initial disturbance in the plasma will steepen and eventually resemble the waves observed at 67P/CG. At some point the nonlinear steepening will be constraint by dissipative effects. Due to the frequency dependent damping, the skewness of the wave event will also be affected. High frequency wave packets as seen in Fig. 6 and at 21P/Giacobini-Zinner (Tsurutani et al., 1987) arise when dispersive effects outweigh dissipative ones and balance the nonlinear steepening.

Nonlinear hyperbolic systems are known to be able to develop shock solutions, which are difficult to treat numerically, especially for methods based on discretizing derivatives directly. Unsuitable numerical schemes may develop non-physical solutions, e.g. in the form of spurious oscillations. Hence, Clawpack, a software suite specifically developed to solve nonlinear conservation laws, balance laws and other first-order hyperbolic partial differential equations, was used (Clawpack Development Team, 2019). To solve systems of nonlinear hyperbolic equations, Clawpack uses a high-resolution wave propagation algorithm (Ketcheson et al., 2013; LeVeque, 2002). The algorithm is based on a finite volume method utilizing Riemann problems to determine the update of the numerical solution. For this study, the Roe approximate Riemann solver (Roe, 1981) and spatial discretization of second-order are used. For time integration a 4th-order strong stability preserving method (Ketcheson et al., 2013) is chosen and at all boundaries of the simulation box, non-reflecting outflow conditions are enforced.





Since the steepened waves in Fig. 1 do not exhibit any apparent periodic behavior, an initial condition of the form of a single
pulse

$$B' = (A-1)\exp\left(-4\ln(2)\frac{(x'-\delta x')}{w}\right)^2 + 1 \tag{14}$$

is chosen. Due to the factor $4\ln(2)$ in the argument of the exponential function, $w$ corresponds to the full width at half
maximum. Following Shukla et al. (2004) the corresponding disturbances in $\rho$, $u$ and $p$ for a fast mode type wave are

$\rho' = B'$ \hfill (15)

$$u' = 2\sqrt{B' + \frac{\gamma\beta}{2}} + \sqrt{\frac{\gamma\beta}{2}}\ln\left(\frac{\sqrt{B' + \frac{\gamma\beta}{2}} - \sqrt{\frac{\gamma\beta}{2}}}{\sqrt{B' + \frac{\gamma\beta}{2}} + \sqrt{\frac{\gamma\beta}{2}}}\right) + c \tag{16}$$

$$p' = \beta\frac{B'^2}{2\mu_0}, \tag{17}$$

with the integration constant

$$c = -2\sqrt{1 + \frac{\gamma\beta}{2}} - \sqrt{\frac{\gamma\beta}{2}}\ln\left(\frac{\sqrt{1 + \frac{\gamma\beta}{2}} - \sqrt{\frac{\gamma\beta}{2}}}{\sqrt{1 + \frac{\gamma\beta}{2}} + \sqrt{\frac{\gamma\beta}{2}}}\right) \tag{18}$$

In the scope of this study we do not explicitly model wave excitation, but instead assume that the initial disturbance is present at
t = 0. Various plasma instabilities triggered by the interaction between the solar wind and newly implanted cometary ions (Wu
and Davidson, 1972; Tsurutani, 1991; Gary, 1991; Motschmann and Glassmeier, 1993; Meier et al., 2016; Glassmeier, 2017)
are known to excite large-amplitude low-frequency waves, which could be the initial disturbance from which such steepened
waves develop. We also want to note that at 26P/Giacobini-Zinner (Tsurutani et al., 1990) and at 19P/Borrelly (Tsurutani
et al., 2013) large-amplitude symmetric pulses, similar to the initial disturbance chosen for the simulation, were found near
the bow shock region. Figure 12 shows the solution to Eq. (9) - (12) computed with Clawpack in the domain $\Omega = [0, 100]$ for
the initial condition given by Eq. (14) to Eq. (17) with the amplitude $A = 2$, width $w = 3$, displacement $\delta x = 30$, $\eta = \nu = 0$,
plasma $\beta = p/p_{mag} = 2c_s^2/(\gamma v_A^2) = 2$ and grid size $\Delta x = 2.5 \times 10^{-2}$. The solid black line denotes the initial condition for the
magnetic field $B$, mass density $\rho$, velocity $u$ and temperature $T$. The dashed line shows the solution after a time t = 15 $\Omega_i^{-1}$.
The nonlinear steepening, as well as the decrease in amplitude, can be observed for all four quantities. A small part of the
initial disturbance can be seen propagating to the left, which is evident in the negative velocity. To ensure that the dissipation
is not numerical, simulations with identical initial conditions but increasing resolution were run. All simulations produced the
same results, independent of the chosen grid size $\Delta x \in \left[0.6 \times 10^{-2}, 5 \times 10^{-2}\right]$.

       In Fig. 13 (a) the steepening time $t_{st}$, which is defined as the time at which the skewness has reached its maximum value,
as a function of the plasma $\beta$ is shown. It decreases with an increasing plasma $\beta$. A similar dependency is observed for the
amplitude of the initial disturbance in Fig. 13 (b). In general, $t_{st}$ declines when the phase velocity of the wave increases. Times
in SI units can be obtained by multiplying $t_{st}$ by the ion gyration time $\Omega_i^{-1}$. For typical values $\Omega_i^{-1} \approx 10$ s/rad we obtain
steepening times in the range between 30 s to 460 s. With typical phase velocities around the order of $v_{ph} \approx \sqrt{v_A^2 + c_s^2} \approx 10$



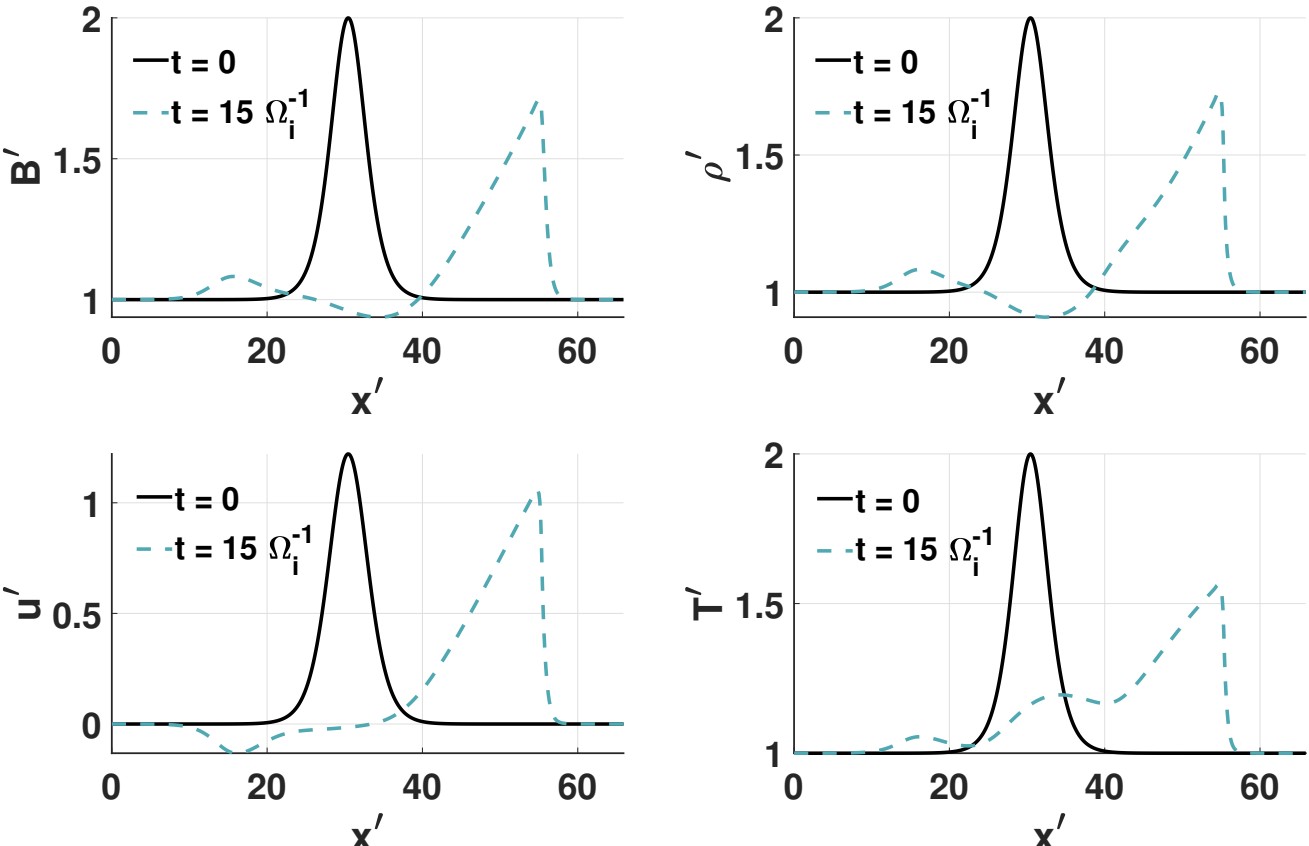

**Figure 12.** Solution of the modified 1D-MHD equation for the initial conditions given by Eq. (14) to (17). The solid black lines shows the initial conditions. The dashed line illustrates the solution computed with clawpack after a time t=15 $\Omega_i^{-1}$.

$\text{kms}^{-1}$, a wave travels between 300 km to 4600 km before reaching its maximum skewness. In both cases, the cometary

interaction region, with an estimated extend ≈ 10000 km, is larger than the distance traveled by the wave. Hence, fast mode waves at 67P/CG have enough time as well as space to fully steepen in the interaction region. Furthermore, it can be assumed that deep in the interaction region, where Rosetta was mostly located, the waves will already have steepened to their maximum skewness. Then the observed skewness of the wave is a function of its amplitude, width and the local plasma properties, in particular the plasma $\beta$, the resistivity and the viscosity.

To obtain an estimate for the effective plasma resistivity and viscosity the relation between skewness, viscosity, resistivity and the plasma parameters needs to be modelled. In Fig. 14 the maximum skewness as a function of viscosity is shown for different amplitudes $A$ (panel a), widths $w$ (panel b) and plasma $\beta$ (panel c). In general, the skewness depends on the ratio between the nonlinear and diffusive terms. Since diffusive terms tend to smooth large gradients, the skewness decreases with increasing viscosity. The strength of the nonlinear term directly depends on the amplitude of the disturbance. Thus, for higher amplitudes

the steepening is more effective than diffusion and higher skewness values can be reached. The behavior for an increase in





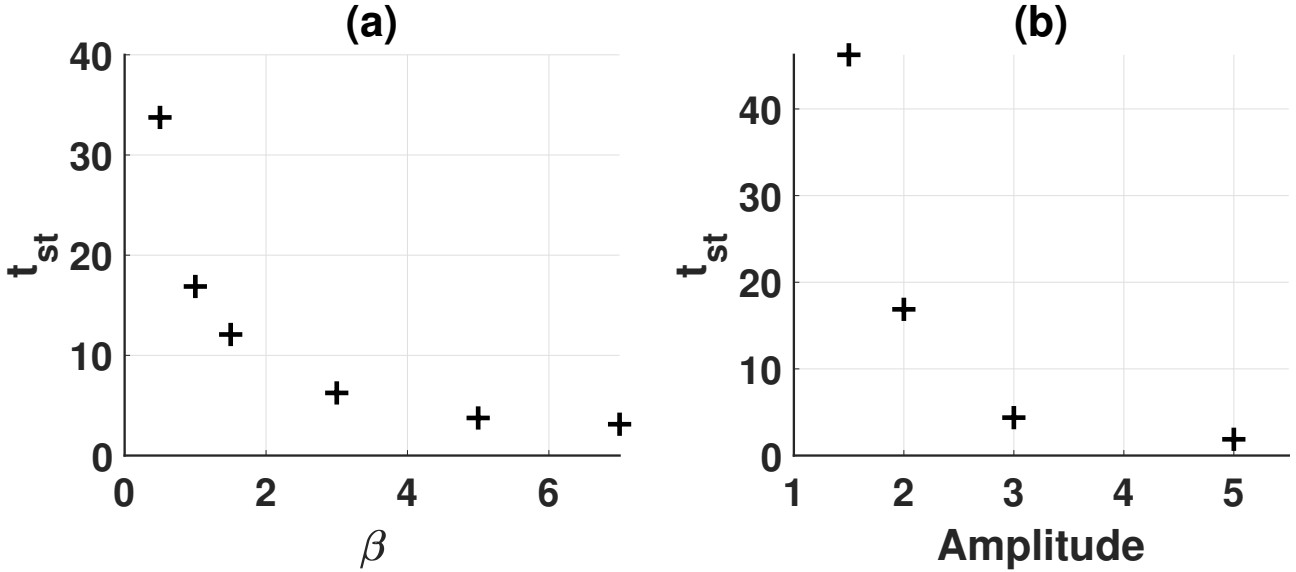

**Figure 13.** Normalized steepening time as a function of the plasma $\beta$ (a) and the amplitude (b).

the width $w$ is similar. The second spatial derivative predominantly dampens structures on small scales. Hence, the damping effect is weaker the larger the structure is. In comparison to the amplitude and width, the viscous term is virtually independent of $\beta$. Both the effectiveness of the non-linear as well es the viscous term increase with the plasma $\beta$. As a consequence, the influence of both terms on the skewness is approximately balanced out. Since the width $w$ of the waves cannot be measured

directly, an approximate value from the temporal width of the wave events and the local magnetosonic speed $v_{ms} = \sqrt{v_a^2 + c_s^2}$ is used for the following simulations. Based on this approximation, the waves have a width of $w = 4$ with a standard deviation of 2 in normalized units. Then, the approximation for the effective viscosity only depends on the skewness and amplitude of the wave. The dependency of the latter is that of a quadratic equation of the form

$$S(A, \nu) = c + p(A)\nu^2, \tag{19}$$

where c = 0.955 is the maximum skewness value given by the SND, $S(A, \nu)$ is the skewness and

$$p(A) = \left( \frac{p_1}{A^2 + p_2 A + p_3} \right) \tag{20}$$

is a parameter depending on the amplitude. The coefficients $p_1 = -0.08$, $p_2 = -2.61$ and $p_3 = 1.79$ are obtained by fitting the value $p(A)$ for different amplitudes. Solving for $\nu$ in Eq. (19) yields the inverse equation

$$\nu_{sim} = \nu(S, A) = \sqrt{\frac{S - c}{p(A)}} \tag{21}$$

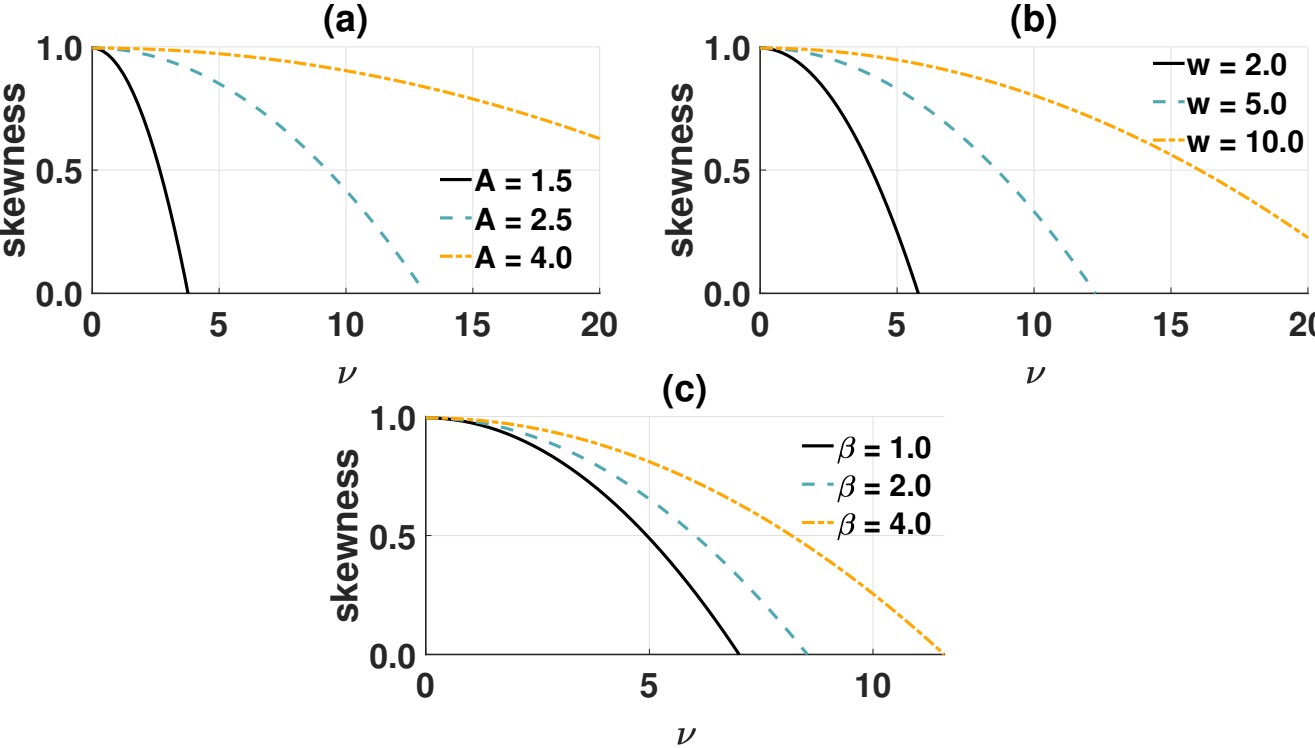

**Figure 14.** Skewness as a function of the kinematic viscosity for different values for the amplitude (a), width (b) and plasma $\beta$ (c).

from which an estimate of the effective viscosity can be obtained if the amplitude $A$ and the skewness $S$ are known. For the effective resistivity additionally the dependency on $\beta$ has to be modeled

$$s(A, \beta, \eta) = c + p(A, \beta)\eta^2, \tag{22}$$

$$p(A, \beta) = \left( \frac{p_1}{A^2 + p_2 A + p_3} \right) \left( \frac{p_4}{\beta^2 + p_5 \beta + p_6} \right). \tag{23}$$

with the coefficients $p_1 = 0.282$, $p_2 = 0.486$, $p_3 = -2.275$, $p_4 = -0.330$, $p_5 = 1.117$ and $p_6 = 0.366$. Solving (23) for $\eta$ yields

$$\eta_{sim} = \eta(S, A, \beta) = \sqrt{\frac{S - c}{p(A, \beta)}}. \tag{24}$$

Over the course of a simulation the amplitude $A$ decreases, while the width $w$ increases due to the presence of diffusive terms. Since it is uncertain where these waves are excited and how far they traveled before Rosetta observed them, no information about the wave properties at the point of excitation are available. Hence, for lack of better information the observed values for the amplitude and width are used in Eq. (21) and (24).



As a quality measure, the computed values are subsequently compared to suitable reference values. For the dynamic viscosity the definition from Khodachenko et al. (2004)

$$\mu_{ros} = \frac{1.92 n_{i,0} k_B T_{i,0}}{\nu_i}, \tag{25}$$

with $\nu_i = \nu_{ii} + \nu_{in} \approx \nu_{in}$, is chosen as a reference. The expression used for the viscosity is highly simplified. More complex
and detailed expression can be found in e.g. Zhdanov (2002). However, for typical conditions at 67P/CG we obtained similar values for Eq. (25) and the expressions given by Zhdanov (2002). Hence, in the following the simpler expression Eq. (25) will be used. The resistivity is governed by electron-neutral collisions

$$\eta_{ros} = \eta_S = \frac{m_e \nu_{en}}{n_e e^2}. \tag{26}$$

Since $\nu_{en} \gg \nu_{ei}$ the contribution through ion-electron collisions was neglected. The collision frequencies are given by

$$\nu_{in} = \sigma_{in} n_n v_i \tag{27}$$

$$\nu_{en} = \sigma_{en} n_n v_e. \tag{28}$$

To facilitate a comparison between computed and reference values, Eq. (25) and Eq. (26) have to be normalized in accordance with Eq. (9) to Eq. (12):

$$\eta'_{ros} = \eta_{ros} \frac{\Omega_i}{\mu_0 v_A^2} \tag{29}$$

$$\nu'_{ros} = \mu_{ros} \frac{\Omega_i}{\rho_0 v_A^2}. \tag{30}$$

Values for the electron temperature $T_e$, the momentum transfer cross section $\sigma_{in}$ and $\sigma_{en}$ and the ion velocities $v_i$ have to be estimated, as no time-resolved measurements for longer periods are available. For the electron temperature a mean value of $T_e = 5$ eV is assumed (Henri et al., 2017; Engelhardt et al., 2018; Hajra et al., 2018b). From Table 5 in Itikawa and Mason (2005) the corresponding momentum transfer cross-section for electron collisions with $H_2O^+$ is obtained $\sigma_{en} = 5 \times 10^{-20} \mathrm{m}^2$.
A mean value of $v_i = 5$ kms$^{-1}$ for the bulk ion velocity is taken from Vigren et al. (2017) and Odelstad et al. (2018). Shortly after ionization the cometary ions have approximately the same temperature as the neutral gas $T_n \approx 180$ K. This was confirmed by Gunell et al. (2017), who studied ion acoustic waves at a weakly active comet 67P/CG (January 2015). However, Gunell et al. (2017) also reported on a heated ion population around $k_B T_i \approx 1$ eV. Such a warm ion population was also observed at 1P/Halley (Schwenn et al., 1988). Haerendel (1987) and Cravens (1987) argued that frictional heating between the ions
and neutrals was responsible for the warm ion population. A similar process is also expected to heat ions at 67P/CG in the strongly active phase. Hence, for the following analysis $k_B T_{i,0} \approx 1$ eV is assumed. For a solar wind primarily mass-loaded with $H_2O^+$ the ion-neutral momentum transfer cross section was estimated to be $\sigma_{in} = 8 \times 10^{-19} \mathrm{m}^2$ (Mendis et al., 1986; Buti and Eviatar, 1989; Hajra et al., 2018b; Mandt et al., 2019). However, as stated by Mendis et al. (1986) and Gunell et al. (2017) the uncertainty of the cross section is significant, as no reliable laboratory measurements are available. Using the values



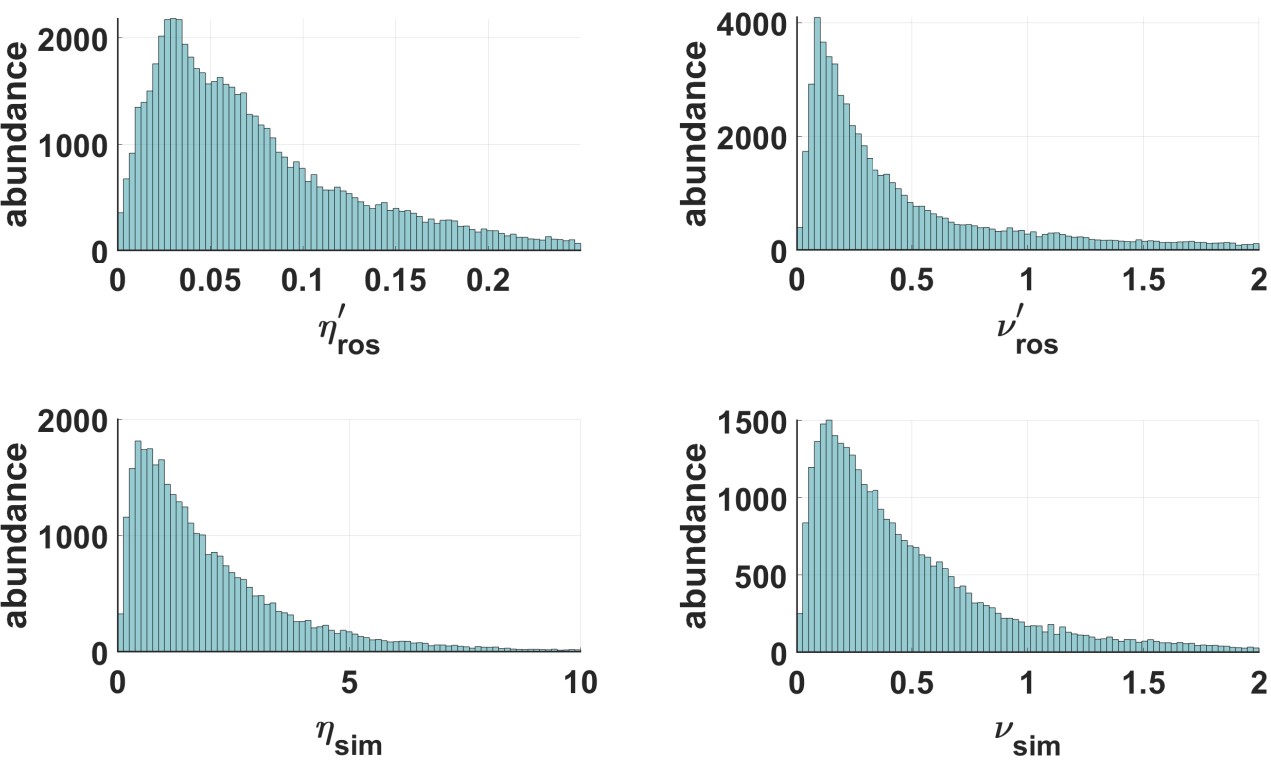

**Figure 15.** Distribution of viscosities and resistivities obtained with simulations ($\nu_{sim}$, $\eta_{sim}$), compared to reference values ($\nu'_{ros}$, $\eta'_{ros}$) computed with Eq. (30) and Eq. (29).

470 given above typical dynamic viscosities and resistivities are $\bar{\mu}_{ros} = 1.76 \times 10^{-9}$ kg/ms, $\bar{\eta}_{ros} = 0.15$ Vm/A and in normalized units $\bar{\nu}'_{ros} = 0.64$, $\bar{\eta}'_{ros} = 4.11 \times 10^{-4}$. The discrepancy between $\bar{\eta}'_{ros}$ and $\bar{\nu}'_{ros}$ amounts to a factor $\sim 1000$. Hence, compared to the viscosity, the resistivity due to electron-neutral collisions is negligible. At high gas production rates collisional cooling of electrons can reduce the electron temperature below 0.1 eV, which is above an order of magnitude lower than the initially assumed 5 eV. For such low temperatures the momentum transfer cross section is $\sigma_{en} \approx 7 \times 10^{-19} \text{m}^2$ (Itikawa and Mason,

475 2005). This yields a mean resisitvity $\bar{\eta}_{ros} = 0.55$ Vm/A, which is slightly larger than the value for the warm electron population but still significantly smaller than the viscosity.

  Figure 15 shows the distribution of $\eta'_{ros}$, $\nu'_{ros}$, $\eta_{sim}$ and $\nu_{sim}$. The shape of the distribution, similar to a Rayleigh distribution, can be reproduced for both cases. The simulated resistivity approximation $\eta_{sim}$ exceeds the reference value $\eta'_{ros}$ by a factor of 100. Investigating diffusion at Earth's magnetopause Nabert (2017) obtained an estimate for the plasma resistivity of $\eta =$

480 $0.4 \times 10^4$ Vm/A. As a typical length scale of the system Nabert (2017) assumed 700 km, which is comparable to the length scales of the steepened waves. The resistivity value given by Nabert (2017) fits the values obtained by our model quite well ($\eta_{sim} \approx 0.3 \times 10^4$ Vm/A), however it is also multiple orders of magnitude higher than the Spitzer resistivity. Even for the high neutral gas densities in the inner coma, the resistivity governed by electron-neutral collisions is to low to explain the observed





variation in the skewness. Hence, electron-neutral collisions likely do not influence the wave damping mechanism. On the other

hand the simulated viscosity approximation $\nu_{sim}$ agrees well with the respective reference values $\nu'_{ros}$. As a second reference, typical values for the dynamic viscosity in the Solar chromosphere, with values for ion temperature, density and neutral density given by Fontenla et al. (1993), lie in the range $10^{-9}$ kg/ms to $10^{-7}$ kg/ms (Vranjes, 2014), which are comparable to the values at 67P/CG. Since the approximated diffusivities from the simulation agree well with the given reference values, the observed wave events can be, on average, described by a combination of nonlinear steepening and a diffusive process balancing said

steepening. Dispersive effects as seen in Fig. 6 (b) are secondary to diffusive effects, as the waves propagate predominantly perpendicular to the background field.

Figure 16 shows time-averaged values of $\nu_{sim}$ and $\nu'_{ros}$ for four different time intervals. Due to the dynamic nature of the interaction region, a comparison is only reasonable on long time scales. Hence, each illustrated interval is between two weeks to one month long. Moreover, a moving average with a window size of two days was applied, which encompasses four

rotational periods of the comet. The model underestimates the viscosity slightly by a factor up to 2. However, the effective viscosity is able to reproduce variations over time, which is indicated by the high correlation coefficients ($\rho_{corr} > 0.7$). In contrast to the resistivity, the viscosity is predominantly influenced by the ion-neutral interaction. Since the variations in the approximated viscosity are sufficiently matched by the reference values, the damping mechanism is likely influenced by the ion-neutral interaction. However, we note that due to temporarily and spatially limited ion temperature and velocity estimations,

the ion-neutral collisionality can only be roughly approximated.

## 8    Conclusion

We present a comprehensive statistical analysis of nonlinear wave phenomena in the inner coma of comet 67P/CG from December 2014 to June 2016 as a general overview of the properties of said wave events. The around 70000 identified events were analyzed to characterize these waves, in particular in relation to the evolution of the cometary interaction region and the

changing plasma conditions in the inner coma of 67P/CG. We observe that the number of detected wave events depends on the local mass-loading. From May 2015 to December 2015 these waves dominate the innermost interaction region with typical times of 5 - 10 minutes between two wave observations. During this period occasional transitions into regions free of wave events within the span of 1 - 2 days were observed. This change of the interaction region is most likely caused by transient solar wind events, which is supported by the observation of a smooth simultaneous increase of the mean magnetic field.

Based on a minimum variance analysis, the wave normal direction and ellipticity were determined. On average the waves are highly elliptically polarized with a mean ellipticity of 0.7. The propagation direction is approximately perpendicular to the background magnetic field, which is typical for fast magnetosonic waves. The compressive nature of the waves is further supported by accompanying enhancements in the electron density (Engelhardt et al., 2018; Hajra et al., 2018b). The waves propagate approximately at an angle of $\pm 35°$ to the comet and at an angle of $\pm 65°$ to the Sun. The pattern the minimum

variance direction exhibits resembles the general ion motion close to the nucleus. Due to the ambiguity of the MVA to the

26





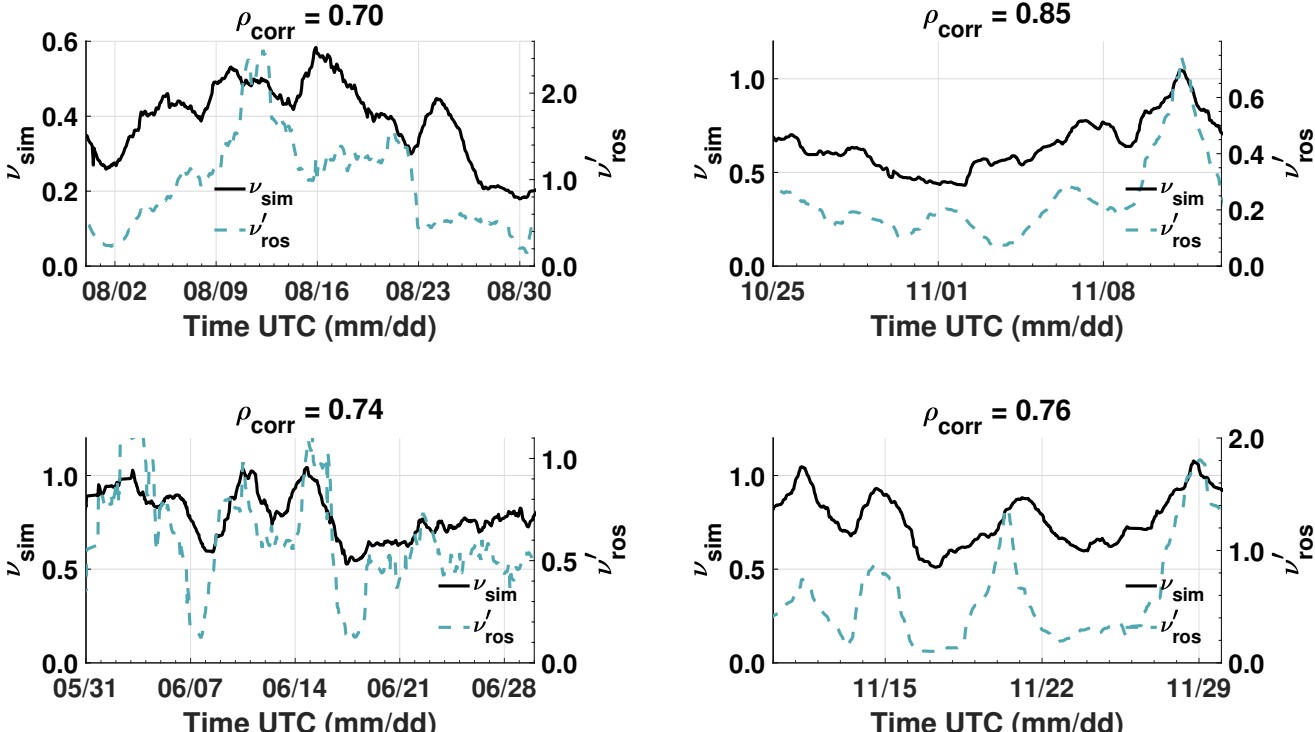

**Figure 16.** Averaged computed kinematic viscosity $\nu_{sim}$ and reference kinematic viscosity $\nu'_{ros}$ over time for four different time intervals in 2015.

orientation, it is unclear if these waves originate from the inner coma and propagate outwards or if they originate from outer regions and propagate inwards.

By fitting a skew normal distribution to the magnetic field magnitude, comprehensive measures for the amplitude, width and skewness of the wave events were obtained. While the skewness increases with rising neutral gas density, the amplitude
decreases and the width shows no apparent correlation. Using a 1D MHD model we showed that the steepened waves are likely nonlinear phase steepened waves balanced by a combination of dispersive and dissipative effects. For average conditions at 67P/CG steepening times are between 30 s to 460 s. With an estimated phase velocity of $v_{ph} = \sqrt{v_A^2 + c_s^2} \approx 10 \text{ km s}^{-1}$ and an approximate extent of 10000 km of the cometary interaction region, the waves have enough time and space to fully steepen in the interaction region. Moreover, we were able to link the observed variation in the waves skewness to a diffusive process
likely influenced by the ion-neutral interaction.

As substantial carriers of energy they actively influence the ambient plasma, e. g. in the form of an additional heating mechanism, with interesting implication for the inner coma. Of particular interest is the interaction of the steepened waves with the diamagnetic cavity boundary and the resulting impact on the cavity properties, which remains a topic for further investigation.



*Code and data availability.* The Rosetta plasma data used in this paper are stored at the PSA archive of ESA (https://archives.esac.esa.int/psa/, PSA, 2020) and the PDS archive of NASA (https://pds.nasa.gov/, PDS, 2020) and are publicly available. A list of Rosetta science events is made publicly available by ESA (https://www.cosmos.esa.int/web/rosetta/science, last access: 5 July 2020. A list of heliosphere current sheet crossings is provided by the Wilcox Solar Observatory and Leif Svalgaard (http://wso.stanford.edu/SB/SB.html, last access: 5 July 2020). Clawpack (http://www.clawpack.org, Clawpack development team, 2020) is distributed under the Berkeley Software Distribution (BSD)
license at https://github.com/clawpack (Clawpack development team, 2020).

*Author contributions.* KO wrote the manuscript and conducted the data analysis. HR participated in adapting and running the simulations. PhH validated the data analysis. KHG, BT, PhH, CG, PiH, MR, IR contributed to the interpretation of the results. All authors contributed to the proofreading of the manuscript.

*Competing interests.* The authors declare that they have no conflict of interest.

*Acknowledgements.* Rosetta is an ESA mission with contributions from its member states and NASA. We are indebted to the whole Rosetta Mission Team, SGS and RMOC for their outstanding efforts in making this mission possible. The RPC-MAG data are made available through the PSA archive of ESA and the PDS archive of NASA.

*Financial support.* The contribution of the RPC-MAG and ROMAP team was financially supported by the German Ministerium für Wirtschaft und Energie and the Deutsches Zentrum für Luft- und Raumfahrt under contract 50QP1401. Research reported in this publication was sup-
ported by the King Abdullah University of Science and Technology (KAUST). Portions of this research were performed at the Jet Propulsion Laboratory, California Institute of Technology under contract with NASA. C.G. was supported by an ESA Research Fellowship. Work at CNRS (LPC2E & Lagrange) is supported by CNES.





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
