# Peer review of "Steepening of magnetosonic waves in the inner coma of comet 67P/Churyumov-Gerasimenko"

_Annales Geophysicae, 2020_

## Referee Comment (RC1) · Anonymous Referee #1 · 12 Jan 2021

This manuscript presents an extensive statistical study of steepened magnetic field structures observed in the environment of comet 67P-CG by the Rosetta spacecraft over the course of its mission. The analysis concentrates on magnetic field measurements, and uses machine learning methods, presented in another paper, to automatically detect the structures. The variation of the occurrence rate of the structures and of their shape as a function of the comet's activity are investigated. Using minimum variance analysis, the authors determine that they propagate roughly in the direction perpendicular to the ambient magnetic field. Because of this transverse propagation, and of the density variations accompanying these structures as reported in other studies, the authors conclude that they are likely magnetosonic in nature. Finally, a 1D

MHD model is used to simulate the non-linear evolution of plasma and magnetic field pulsations into steepened structures.

This paper is well-written and contains interesting new results. It lacks however a more detailed comparison with previous works published on very similar observations by Rosetta, and more discussion about the physical processes at play and the possible sources of the waves. I have listed my comments below, first these major points, then minor comments. The manuscript should be suitable for publication in Annales Geophysicae after these are addressed.

Major comments:

Introduction: It would be interesting to include more information in the introduction regarding the sources of low-frequency waves in cometary environments, which are relevant to the present study. At the moment, it is only mentioned at lines 386-387 that waves can be triggered by "various plasma instabilities", while the survey of low-frequency wave observations at comets provided at lines 32-39 in the introduction only reports on the wave properties, and not their inferred sources and the physical processes at play.

I would also recommend to include more details on the studies by Engelhardt et al. [2018] and Hajra et al. [2018b], which report on similar structures observed by Rosetta, the former outside of the diamagnetic cavity and the latter inside the cavity. It is worth noting for example that Engelhard et al. find that the duration of the structures are typically a few second to a few tens of seconds, and that their occurrence rate is largest near perihelion, consistent with the present study. This would allow to better place the present study in the context of the existing literature. Furthermore, the comparison with these previous studies raises some interesting questions, in particular regarding the nature of the steepened magnetic field signatures. Both Engelhardt et al. and Hajra et al. conclude that the observed signatures are more likely to be plasma structures rather than waves, which are associated with the boundary of the diamagnetic cavity (either

filamentary structures extending from the diamagnetic cavity, or structures inside the cavity caused by disturbances at its boundary). In contrast, in the present study, it is assumed that the observed magnetic structures are steepened fast-magnetosonic waves. More discussion is needed to reconcile these different assumptions regarding the nature of these steepened magnetic field signatures, either already in the introduction, since the authors then rely on this assumption to estimate the velocity of the waves later in the study, or in the conclusion section.

Another aspect of the study which, in my opinion, requires more discussion is the selection of the events. The description of the event selection method is rather brief, as the authors have published a more detailed presentation of their method in another paper, but there are two points that would deserve some clarification in the present manuscript: First, was the position of Rosetta within the cometary environment taken into account when selecting the events? In particular, were the intervals when the spacecraft was within the diamagnetic cavity excluded from the analysis? Or was it assumed that there would be no detection of steepened waves inside this cavity, since the magnetic field is almost zero in this region? Second, and more importantly, the study by Hajra et al. [2018b] reveals that the crossings of the boundary of the diamagnetic cavity display signatures that are extremely similar to those of the steepened waves presented here (see their Figures 1 and 2). Would these crossings be picked up by the detection method used in the present study? If so, how would this affect the analysis and the conclusions?

The study by Engelhardt et al. [2018], which focused on similar structures but taking into account plasma measurements in addition to magnetic field data, concluded that the occurrence of these structures strongly depended on the distance from the electron exobase. Would it be possible to calculate this parameter for the present, much more extended, data set, or at least for part of it when the relevant data are available, and check whether a similar distribution of the observations is found?

Minor comments:

Page 2, line 25 "In such regions, conditions for the steepening of compressive modes are exceptionally favourable." Can the authors briefly explain why this is the case for cometary interaction regions?

Figure 1: I would suggest to add one plasma parameter, for example an ion time-energy spectrogram, to showcase that the spacecraft remains in the same plasma region, and that the non-linear enhancements are indeed waves rather than boundaries between plasma regions.

Page 6, lines 111-112: "The exact nature of this transition region and the processes governing it require a more in-depth analysis, which is out of the scope of this paper." I am not sure I understand which transition region is referred to here. Is it the transition between the magnetic cavity and the outside solar wind, or the transition from low-activity cometary environment (with the "singing comet" waves) to high-activity cometary environment (with steepened waves)? Could you please clarify?

Page 7 and Figure 3: the occurrence rate shows a plateau above a certain massloading rate. It would be interesting to add more discussion about what could cause that. Is this physical, or could this be due to errors in the measured neutral gas density? Or could it be that the ionisation rate is incorrectly modelled above a certain activity level?

Page 7, lines 157-159: "In some cases, these sharp increases coincide with increases in the solar wind dynamic pressure. However, most of the time, no correlation between the pressure and time between observations is visible." It is not clear to me why it would be expected that higher dynamic pressure would lead to larger time between the wave observations. Longer intervals suggests a lower wave activity in the comet's environment, whereas high dynamic pressure rather corresponds to "disturbed" solar wind conditions. Or is it because the comet's environment would be compressed, resulting in the spacecraft being located in a different environment? Could the authors please elaborate on this point?

Section 4: Similar dispersive wave signatures are also observed in association with steepened waves in the Earth's foreshock [e.g. Hada, T., C. F. Kennel, and T. Terasawa: 1987, 'Excitation of compressional waves and the formation of shocklets in the Earth's foreshock'. J. Geophys. Res. 92(5), 4423–4435 and Greenstadt, E. W., G. Le, and R. J. Strangeway: 1995, 'ULF waves in the foreshock'. Adv. Space. Res. 15, 71–84]. How do they compare with the observations reported in the present manuscript?

Section 5: Is the fitting applied to all waves, or only to those that do not show dispersive effects? Does it affect the results?

Page 12, lines 232-233: Does the data set include waves with negative skewness, which were discarded due to the constraint on the skewness being > 0.6?

Page 13 and Figure 8: It would be interesting to discuss whether the algorithm could affect the final distribution of skewness. In particuar, does the algorithm detect more efficiently highly-steepened waves, thus introducing a possible bias?

Page 15, lines 264-265: The authors state that the waves are well-defined for a ratio > 13.7. Could you please provide a reference for this threshold? Or is this an observation made from the present analysis?

Page 17, point 1: "Influences of extreme solar wind conditions can be seen in occasional sudden increases of the time between observation of two events." In my opinion, the present study does not provide sufficiently convincing evidence of the influence of extreme solar wind conditions to support this statement. I would suggest to tone it down to be more in line with the findings of the present work ("may be seen" instead of "can be seen" for example). Also, as it is now, it contradicts what is stated in the abstract (at lines 8-9).

Figure 15 and associated text: I am not sure I understand how the parameters displayed in Figure 15 are obtained. Did the authors run their model for all steepened wave observations in their data set? And similarly, did they calculate the associated

values for the resistivity and viscosity based on Eq 25 and 26 for each interval, using plasma observations made simultaneously with the detection of steepened magnetic field structures?

Lines 508-509: "This change of the interaction region is most likely caused by transient solar wind events, which is supported by the observation of a smooth simultaneous increase of the mean magnetic field." According to Figure 5, an increase of the background magnetic field is observed during thoses intervals, rather than the "smooth increase" described here, which reads as if the field strength changes progressively over the course of the event. I would suggest to reformulate this. It could also be interesting to add to the discussion that measurements from SREM could provide additional information regarding the solar wind conditions, and Enlil simulations could show whether transient solar events may be reaching the comet at these times (see for example the study by Witasse et al., 2017, "Interplanetary coronal mass ejection observed at STEREO-A, Mars, comet 67P/Churyumov-Gerasimenko, Saturn, and New Horizons en-route to Pluto. Comparison of its Forbush decreases at 1.4, 3.1 and 9.9 AU", doi:10.1002/2017JA023884). This additional analysis lies of course beyond the scope of the present study, but it'd be worth mentioning.

Lines 514-515: "The pattern the minimum variance direction exhibits resembles the general ion motion close to the nucleus." It would be interesting to discuss what are the possible implications of this finding regarding the source of the waves. Would this hint at one of the instabilities mentioned in previous studies?

Lines 519-520: "While the skewness increases with rising neutral gas density, the amplitude decreases" Would it be possible to distinguish between the increase in neutral gas density due to cometary activity and that due to the distance from the comet? If so, could this help in identify the source region of the waves, assuming that the skewness increases as the waves evolve and steepen with time?

Technical corrections:

Page 6, line 115: "implies" -> "is associated with" ("implies" suggests that there's a direct causation, which cannot be established on the sole basis of the present study)

Figure 10: The right-hand side of the right panel seems to be cut: there's a few bins missing to reach 180 degrees, with only a vertical line remaining around 170 degrees.

Page 15, lines 273-274: "the following analysis was only performed for the periods in which diamagnetic cavities were available to adjust the offsets." → "the following analysis was only performed for the periods in which observations of the diamagnetic cavity were available to adjust the offsets."

Page 17, point 4: "at an angle to the Sun" -> to the Sun-comet line?  to the Sun-spacecraft line?

Line 418: "es" -> "as"

Eq 22: "s" should be "S"

Line 471: "discrepancy" reads as if this difference between the two parameters is an error, whereas it is actually an observation that is made here, based on the model, unless I am mistaken. I would suggest to reformulate this sentence, for example "$\eta$ and $\nu$ differ by a factor of $\sim 1000$"

Line 483: "to low" -> "too low"

Lines 507-508: "During this period occasional transitions into regions free of wave events within the span of 1 - 2 days were observed." This sentence reads as if Rosetta was probing a different part of the cometary environment, which didn't have such steepened waves, during these intervals.  However, based on the presented analysis (and the next sentence) it is rather that the waves "disappear" from the cometary environment during these intervals.  I would suggest to reformulate this sentence to better convey this.

Line 515: Again, the angles "to the comet" and "to the Sun" should be rather to the

"comet-spacecraft line" and to the "Sun-comet" or "Sun-spacecraft" line to be unambiguously defined.
* * *

---

## Referee Comment (RC2) · Martin Volwerk (Referee) · 13 Jan 2021

Referee Report Ostzszevski et al. Title: Steepening of magnetosonic waves in tne inner coma of comet 67P?Churyumov-Gerasimenko

This paper deals with the very interesting topic of so-called "steepened waves" that have been observed in the Rosetta magnetometer data, in the inner coma of comet 67P. These objects also have a corresponding signature in the plasma data (even in the diamagnetic cavity, i.e. without a magn0etic field), and therefore it is necessary to understand the characteristics of these waves. The authors make a thorough investigation of the data, where they show the various details of the waves in a statistical

way. They find that the waves travel almost perpendicular to the magnetic field and are thus most-likely fast-mode waves. Unfortunately, the plasma velocity vector cannot be determined, and thus, using minimum variance analysis, this leaves a sign ambiguity. The authors then use a 1D MHD description to model these waves and compare these results with the characteristics determined from the data.

This paper is well written, rather long (maybe a 2-papers version would have been an idea?), and goes deep into the material. It is definitely a great resource for further studies of these waves. There are some (mostly minor) comments that I have listed below.

Line 50: It would be nice to give the date of perihelion, so "(13 August 2015)" instead of "(August)"

Line 78: Is it really necessary to cite Glassmeier et al. a through g? I do not see much use in the references to these PSA documents.

Line 132: forgotten space between distance and (Biermann

Line 137: we use THE locally . . .

Line 147: The authors state that "after which the detection rate stagnates" referring to figure 3, after a mass-loading M > 2 kg km-3 s-1. Another interpretation could be that the detection rate does not "stagnate", which can imply that Rosetta would not be able to measure more waves for some reason, but that the generation mechanism (which is not discussed in the paper) saturates. That somehow, above a certain mass-loading rate the generation of "solitons" (?? Like the input in the numerical part later in the paper) reaches a limit. Of course, finding the source for the steepened waves is rather difficult with the limited data that Rosetta delivered, if it is actually "solitons" or not, but those waves do steepen as shown later in the paper.

Line 167: The authors here discuss the direction of the field, which only slightly changes, and then two magnetic vectors are shown. This is difficult to interpret, I

would rather see either normalized vectors or angles.

Line 189: The authors have looked at differences in cometary outgassing activity to see how that influences the detection of the steepened waves. Later, they find differences in the correlation between mass-loading and skewness and amplitude. In the modeling section the authors assume a pure mass 18 plasma. However, we also know that CO and $CO_2$ are major components in the outgassing, also depending on which hemisphere is more active. Indeed, Heritier et al (2017) say that the cops instrument is less sensitive for $CO_2$ than for the water group, but it still is a significant species. Figure 8: In order to be able to really compare the top and bottom rows of plots, the Y-axes should be normalized by total number of points in each row.

Line 279: forgotten "," after Narita (2017)

Line 420: Here I am not sure if the authors have looked at this or not. From the simulations the width of the waves is determined and in the simulation the velocity is also know. Thus one could calculate the duration in seconds of these steepened waves and compare them with the observed width in seconds.

Line 444: A More complex . . .

Line 473: "above" should read "more than"

Line 475: Here I do not understand the comparison. The authors write: "This yields a main resistivity eta, which is slightly larger than the value for the warm electron population but still significantly smaller than the viscosity." How do the authors compare values of completely different units, resistivity and viscosity, and then determine which is "smaller"?

Line 514: at an angle of pm 35 to the comet and at an angle of pm 65 to the sun. I think here some more direction information is needed, than just these angles, e.g. "to the comet-rosetta direction" or the "sun-rosette/comet direction".

---

## Author Comment (AC1) · 23 Feb 2021

**Response to review by Martin Volwerk (Referee #2)**

We wish to thank Martin Volwerk for his valuable input and evaluation of our manuscript. Below, we have included the referee comments in italics and our own response in regular text.

*This paper deals with the very interesting topic of so-called "steepened waves" that have been observed in the Rosetta magnetometer data, in the inner coma of comet 67P. These objects also have a corresponding signature in the plasma data (even in the diamagnetic cavity, i.e. without a magnetic field), and therefore it is necessary to understand the characteristics of these waves. The authors make a thorough investigation of the data, where they show the various details of the waves in a statistical way. They find that the waves travel almost perpendicular to the magnetic field and are thus most-likely fast-mode waves. Unfortunately, the plasma velocity vector cannot be determined, and thus, using minimum variance analysis, this leaves a sign ambiguity. The authors then use a 1D MHD description to model these waves and compare these results with the characteristics determined from the data. This paper is well written, rather long (maybe a 2-papers version would have been an idea?), and goes deep into the material. It is definitely a great resource for further studies of these waves. There are some (mostly minor) comments that I have listed below.*

*It would be nice to give the date of perihelion, so "(13 August 2015)" instead of "(August)"*
In lieu of changes made in regard to comments from referee #1 the reference to the date of perihelion was removed. The paragraph now reads: "Figure Fig. 1 shows two exemplary intervals with multiple of these waves in the magnetic field data and electron density on 16 July 2015 and 21 November

2015. During both intervals the outgassing rate was already high enough to facilitate the development of a diamagnetic cavity (Goetz et al. 2016a, Goetz et al. 2016b) and, with a high probability, a bow shock (Koenders et al. 2013, Koenders et al. 2015). The striking features of Fig. 1 are the asymmetric, large amplitude enhancements in the magnetic field and electron density. While properties like amplitude, width and strength of asymmetry can change significantly from event to event, they are still strikingly similar in respect to their general shape. In particular, for all instances of magnetic field enhancements, concurrent enhancements in the electron density are visible.

*Is it really necessary to cite Glassmeier et al. a through g? I do not see much use in the references to these PSA documents.*
We removed the citations.

*forgotten space between distance and (Biermann*
We added the space.

*we use THE locally*
Corrected.

*The authors state that "after which the detection rate stagnates" referring to figure 3, after a mass-loading M >2 kg km-3 s-1. Another interpretation could be that the detection rate does not "stagnate", which can imply that Rosetta would not be able to measure more waves for some reason, but that the generation mechanism (which is not discussed in the paper) saturates. That somehow, above a certain mass-loading rate the generation of "solitons" (?? Like the input in the numerical part later in the paper) reaches a limit. Of course, finding the source for the steepened waves is rather difficult with the limited data that Rosetta delivered, if it is actually "solitons" or not, but those waves do steepen as shown later in the paper.*
We have added a more thorough discussion about possible explanations for the plateau in the detection rate also following remarks from referee #1 (see response to referee #1).

*The authors here discuss the direction of the field, which only slightly changes, and then two magnetic vectors are shown. This is difficult to interpret, I would rather see either normalized vectors or angles.*

We have normalized the magnetic field vectors as suggested.

*The authors have looked at differences in cometary outgassing activity to see how that influences the detection of the steepened waves. Later, they find differences in the correlation between mass-loading and skewness and amplitude. In the modeling section the authors assume a pure mass 18 plasma. However, we also know that CO and CO2 are major components in the outgassing, also depending on which hemisphere is more active. Indeed, Heritier et al (2017) say that the cops instrument is less sensitive for CO2 than for the water group, but it still is a significant species. Figure 8: In order to be able to really compare the top and bottom rows of plots, the Y-axes should be normalized by total number of points in each row.*

While CO and $CO_2$ are major outgassing species, the period we consider is heavily water dominated (see e. g. Läuter, Matthias and Kramer, Tobias and Rubin, Martin and Altwegg, Kathrin, The gas production of 14 species from comet 67P/Churyumov–Gerasimenko based on DFMS/COPS data from 2014 to 2016, MNRAS 498 , pages 3995-4004, 2020, 10.1093/mnras/staa2643). Hence, we expect the effect of not taking the full composition into account to be less than 10 %. The Y-axes were normalized as suggested.

*forgotten "," after Narita (2017)*
We have added the ",".

*Here I am not sure if the authors have looked at this or not. From the simulations the width of the waves is determined and in the simulation the velocity is also know. Thus one could calculate the duration in seconds of these steepened waves and compare them with the observed width in seconds.*

We have computed the width of the waves in seconds as suggested and obtain values in the region of 10s to 150s, which are comparable to the durations observed by Rosetta. We have added the following remark in the manuscript: "As a consistency check we computed the temporal width of the simulated waves at different times in the simulation and compared them with observed temporal widths of the waves (Fig. 8). Depending on the parameters for the initial condition and $\Omega_i^{-1} \approx 10$ s/rad we obtained durations between 10 s up to 150 s, which is consistent with the observed durations."

*A More complex ...*
We have added the "A".

*"above" should read "more than"*
We replaced "above" with "more than" as suggested.

*Here I do not understand the comparison. The authors write: "This yields a main resistivity eta, which is slightly larger than the value for the warm electron population but still significantly smaller than the viscosity." How do the authors compare values of completely different units, resistivity and viscosity, and then determine which is "smaller"?*
The comparison was meant in regards to the normalized values, which was indeed not clearly stated in the sentence. We have now changed the sentence to: "This yields a mean resistivity $\eta_{ros} = 0.55$ Vm/A, which is slightly larger than the value for the warm electron population. However, in normalized units the resistivity for the cold electron case is still significantly smaller than the viscosity."

*at an angle of pm 35 to the comet and at an angle of pm 65 to the sun. I think here some more direction information is needed, than just these angles, e.g. "to the comet-rosetta direction" or the "sun-rosette/comet direction".*
We have added the additional information as suggested.

---

## Author Comment (AC2) · 23 Feb 2021

**Response to review by anonymous referee #1**

We wish to thank the anonymous referee for his valuable input and evaluation of our manuscript. Below, we have included the referee comments in italics and our own response in regular text.

**Major comments:**

It would be interesting to include more information in the introduction regarding the sources of low-frequency waves in cometary environments, which are relevant to the present study. At the moment, it is only mentioned at lines 386-387 that waves can be triggered by "various plasma instabilities", while the survey of low frequency wave observations at comets provided at lines 32-39 in the introduction only reports on the wave properties, and not their inferred sources and the physical processes at play.

We have added the following paragraph in the introduction: "As source mechanisms for these waves ion ring-beam instabilities (Wu et al. 1972) and non-gyrotropic phase space density-driven instabilities (Motschmann et al. 1993) were proposed. Depending on the angle between the interplanetary magnetic field  $B_{IMF}$  and the solar wind flow  $u_{sw}$  newborn cometary ions form ring, ring-beam or beam distributions in velocity space. In general, these distributions are unstable to resonant wave growth (Sagdeev et al. 1986, Gary et al. 1991). A full ring-beam distribution is only formed if cometary ions are produced over lengths scales larger than the cometary ion gyroradius. If ionization takes place on smaller scales, the ring-beam distribution can only be partially filled, leading to a non-gyrotropic distribution as observed at 26P/Grigg-Skjellerup. At 67P/Churyumov-Gerasimenko (67P/CG) a ring-beam distribution was expected at large distances from the comet during the strongly active months around perihelion. However, due to its operational design, Rosetta was primarily located in the innermost interaction region

and, hence, never able to observe a bow shock crossing. Consequently, the existence of nonlinear waves near the bow shock region at 67P/CG is unconfirmed. Such low frequency waves were also expected at lower outgassing rates, but were never observed (Glassmeier 2017). Instead a new type of nonlinear low-frequency waves, often termed singing comet waves, was encountered (Richter et al. 2015). Since the characteristics of these waves did not fit a ring-beam instability, Richter et al. (2015) suggested a cross-field current instability as the possible source mechanism. Based on these observations Meier et al. (2016) proposed a modified ion-Weibel instability."

I would also recommend to include more details on the studies by Engelhardt et al. [2018] and Hajra et al. [2018b], which report on similar structures observed by Rosetta, the former outside of the diamagnetic cavity and the latter inside the cavity. It is worth noting for example that Engelhard et al. find that the duration of the structures are typically a few second to a few tens of seconds, and that their occurrence rate is largest near perihelion, consistent with the present study. This would allow to better place the present study in the context of the existing literature. Furthermore, the comparison with these previous studies raises some interesting questions, in particular regarding the nature of the steepened magnetic field signatures. Both Engelhardt et al. and Hajra et al. conclude that the observed signatures are more likely to be plasma structures rather than waves, which are associated with the boundary of the diamagnetic cavity (either filamentary structures extending from the diamagnetic cavity, or structures inside the cavity caused by disturbances at its boundary). In contrast, in the present study, it is assumed that the observed magnetic structures are steepened fast-magnetosonic waves. More discussion is needed to reconcile these different assumptions regarding the nature of these steepened magnetic field signatures, either already in the introduction, since the authors then rely on this assumption to estimate the velocity of the waves later in the study, or in the conclusion section.

We have included more details on the studies by Hajra et al. (2018b) and Engelhardt et al. (2018) in the introduction, the observations section and the conclusion. While Hajra et al. (2018b) studied unmagnatized density pulses inside the diamagnetic cavity, this study focuses on magnetic field enhancements outside the cavity. We think that these are two completely different things which should not be mixed. The authors argue that the density pulses are likely inward (toward the comet) propagating structures originating from the cavity boundary. One hypothesis by Hajra et al. is that inward propagating magnetized structures cause the density pulses inside the diamagnatic cavity by interaction with the cavity boundary. However, the authors also mention that there is not enough observational evidence to support this hypothesis. We do not think that the interpretation by Hajra et al. contradicts the assumption that the magnetized structures are magnetosonic in nature in any way and have added respective remarks in the introduction.

Engelhardt et al. (2018) note that the density pulses coincide with magnetic field enhancements, which is a clear indication that these structures are magnetosonic in nature. The coincidence of magnetic field and density enhancements results in a pressure imbalance which either causes these structures to propagate like waves or it causes them to disperse. We have modified Fig. 1 in the manuscript to showcasing the concurrent enhancements in the magnetic field and electron density to emphasize this. Moreover structures with similar properties and magnetic field signatures, which were identified to be nonlinear waves, have been observed before in the cometary environment (Tsurutani et al. 1989, etc.) just not in the inner coma. Based on this we think it is also valid to explore the possibility that the structures are waves.

To better convey this we restructured the paper. We briefly discuss both possibilities in the introduction with mentions to the studies by Hajra et al. (2018b) and Engelhardt et al. (2018) and previous observations of steepened waves at comets. Based on the arguments presented above we interpret the plasma and magnetic field pulses as steepened magnetosonic waves. To avoid confusion in nomenclature we then refer to the magnetic field enhancements as magnetosonic waves throughout the study. However, we also want to note that the chosen interpretation does not affect the results of the statistical study presented in Sect. 3 to Sect. 6. At the beginning of Sect. 7 we discuss in more detail additional indications presented in Sect. 3 to Sect. 6 for these structures being waves and how we intend to characterize them by assuming an initial wave-like disturbance and modeling its evolution and properties using a 1D MHD model.

Another aspect of the study which, in my opinion, requires more discussion is the selection of the events. The description of the event selection method is rather brief, as the authors have published a more detailed presentation of their method in another paper, but there are two points that would deserve some clarification in the present manuscript: First, was the position of Rosetta within the cometary environment taken into account when selecting the events? In particular, were the intervals when the spacecraft was within the diamagnetic cavity excluded from the analysis? Or was it assumed that there would be no detection of steepened waves inside this cavity, since the magnetic field is almost zero in this region? Second, and more importantly, the study by Hajra et al. [2018b] reveals that the crossings of the boundary of the diamagnetic cavity display signatures that are extremely similar to those of the steepened waves presented here (see their Figures 1 and 2). Would these crossings be picked up by the detection method used in the present study? If so, how would this affect the analysis and the conclusions?

Due to different sampling rates of the RPC instruments and limited data availability we only used magnetic field data for the event identification. The position of Rosetta was not directly taken into account, however, since the magnetic field inside the cavity is close to zero, the neural network is not able to pick up on wave events inside the cavity. Moreover, since the plasma pulses inside the cavity differ from those outside the cavity by the fundamental fact that they are not accompanied by concurrent enhancements in the magnetic field, they are completely different in nature and should, hence, not be mixed. For this study our focus is on the magnetized structures outside the cavity, therefore we have explicitly excluded events inside the cavity. Even if we would have included events inside the cavity, compared to the number of events outside the cavity ( $\sim 45000$ ), the number of events inside the cavity (probably  $\sim 100$ , 23 in Hajra et al. 2018b) are significantly less and would not impact the statistics.

Crossings of the boundary of the diamagnetic cavity as discussed by Hajra et al. (2018b) were also explicitly excluded from the analysis. It is still unclear if the signatures displayed at the crossings of the diamagnetic cavity boundary are features of the cavity or if they are steepened waves overlapping the cavity boundary. Hajra et al. (2018b) also presented the hypothesis that the density pulses could be caused by the interaction of inward propagating magnetized structures with the cavity boundary. Either way it is difficult to separate the cavity and the steepened waves in these instances, hence we excluded them from the analysis.

The study by Engelhardt et al. [2018], which focused on similar structures but taking into account plasma measurements in addition to magnetic field data, concluded that the occurrence of these structures strongly depended on the distance from the electron exobase. Would it be possible to calculate this parameter for the present, much more extended, data set, or at least for part of it when the relevant data are available, and check whether a similar distribution of the observations is found?

We have added a figure showcasing the occurrence rate of these structures depending on the electron exobase. We find that the distribution is similar to the one presented by Engelhardt et al. (2018), where more structures can be observed closer to the electron exobase.

**minor comments**

Page 2, line 25 "In such regions, conditions for the steepening of compressive modes are exceptionally favourable." Can the authors briefly explain why this is the case for cometary interaction regions?

The effectiveness of compressive wave steepening heavily depends on the plasma beta, where a high plasma beta facilitates quicker wave steepening. However, we realized that the statement is not true in general and hence removed it.

Figure 1: I would suggest to add one plasma parameter, for example an ion time-energy spectrogram, to showcase that the spacecraft remains in the same plasma region, and that the non-linear enhancements are indeed waves rather than boundaries between plasma regions.

We have added electron density observations to showcase that the magnetic field enhancements are accompanied by concurrent density enhancements.

Page 6, lines 111-112: "The exact nature of this transition region and the processes governing it require a more in-depth analysis, which is out of the scope of this paper." I am not sure I understand which transition region is referred to here. Is it the transition between the magnetic cavity and the outside solar wind, or the transition from low-activity cometary environment (with the "singing comet" waves) to high-activity cometary environment (with steepened waves)? Could you please clarify?

The statement was made in regard to the transition between the low-activity cometary environment, in which the singing comet is a predominant feature, to the high activity cometary environment with steepened waves. With rising activity the singing comet waves cannot be observed at some point. It is still unclear if they simply vanish because, e. g. the generation mechanism is inefficient for higher outgassing rates, or if they are obscured by variations with higher amplitudes. We have rephrased the sentence as follows in the hopes that it more clear now: "The exact nature of the transition between the "singing comet" dominated magnetic field to the steepened waves dominated field and the processes governing it require a more in-depth analysis, which is out of the scope of this paper."

Page 7 and Figure 3: the occurrence rate shows a plateau above a certain massloading rate. It would be interesting to add more discussion about what could cause that. Is this physical, or could this be due to errors in the measured neutral gas density? Or could it be that the ionisation rate is incorrectly modelled above a certain activity level?

Also following comments from Martin Volwerk (referee # 2) we have added the following discussion to the manuscript: "As noted by Bieler et al. (2015) the gas production rate at 67P/CG is dominated by H2O, CO2 and CO. While the period considered in this study is heavily water dominated (Läuter et al. 2020), CO2 and CO have a higher molecular mass. Hence, the observed stagnation may be caused by underestimating the local mass-loading. Another possibility is that the observed plateau in the occurrence rate is caused by a saturation of the waves' generation mechanism. Ultimately, a more detailed investigation is needed to resolve this question.

Page 7, lines 157-159: "In some cases, these sharp increases coincide with increases in the solar wind dynamic pressure. However, most of the time, no correlation between the pressure and time between observations is visible." It is not clear to me why it would be expected that higher dynamic pressure would lead to larger time between the wave observations. Longer intervals suggests a lower wave activity in the comet's environment, whereas high dynamic pressure rather corresponds to "disturbed" solar wind conditions. Or is it because the comet's environment would be compressed, resulting in the spacecraft being located in a different environment? Could the authors please elaborate on this point?

As reported by Goetz et al. (2017) increases in the solar wind dynamical pressure primarily affect the magnetic field strength and not the waves activity. The intervals in which the time between wave observations increases rapidly also show a significant increase in the mean magnetic field strength, which can be seen as an indicator for an increase in the solar wind pressure. It is possible that due to the increase in the solar wind dynamical pressure the plasma parameters change in such a way that the evolution of such steepened waves is impeded. Another possibility is that the interaction region is compressed to such a degree that the waves do not have enough time and space to evolve.

Section 4: Similar dispersive wave signatures are also observed in association with steepened waves in the Earth's foreshock [e.g. Hada, T., C. F. Kennel, and T. Terasawa: 1987, 'Excitation of compressional waves and the formation of shocklets in the Earth's foreshock'. J. Geophys. Res. 92(5), 4423-4435 and Greenstadt, E. W., G. Le, and R. J. Strangeway: 1995, 'ULF waves in the foreshock'. Adv. Space. Res. 15, 71-84]. How do they compare with the observations reported in the present manuscript?

The discrete waves packets mentioned in Hada et al. (1987) and Greenstadt et al. (1995) are significantly more pronounced than the observed dispersive effects at 67P/CG. This can be due to the fact that dispersive effects outweigh diffusive effects in Earth's foreshock while at 67P/CG diffusive effects are more important. Similar to the wave packets at 67P/CG the ones described in Hada et al. (1987) and Greenstadt et al. (1995) also occur at the steep edge of asymmetric magnetic field enhancements, referred to as shocklets by Greenstadt et al. (1995), and the amplitude decreases with distance to the steep edge. Based on this it is fair to assume that the physical processes at play for both situations are similar. We have added citations in Section 4 to the mentioned studies.

**Section 5: Is the fitting applied to all waves, or only to those that do not show dispersive effects? Does it affect the results?**

The fitting is applied to all waves. While a significant portion of events shows evidence of dispersive effects in most cases the effects are only mildly pronounced, less than the example given in Fig. 6. Events which show strong dispersive effects with multiple oscillations are excluded from the analysis by the constraint on the goodness of fit.

Does the data set include waves with negative skewness, which were discarded due to the constraint on the skewness being > 0.6?

The data set does not include waves with negative skewness. We have also trained a neural network to detect waves with negative skewness, however it was only able to detect very few events (< 10). Additionally, we tried to manually look for waves with negative skewness for a couple of randomly chosen 1-day intervals during high cometary activity. Again, we only found a couple instances and hence disregard events with negative skewness for this study.

Page 13 and Figure 8: It would be interesting to discuss whether the algorithm could affect the final distribution of skewness. In particular, does the algorithm detect more efficiently highly-steepened waves, thus introducing a possible bias?

We explicitly designed the training data set to contain and even distribution of skewness values so as not to introduce a bias. The same approach was employed in regard to amplitude and width of the waves. We also checked the event selection manually for randomly selected intervals and did not notice a bias. Moreover, the number of identified events decreases very smoothly for smaller skewness values. Consequently, a possible bias would have to reflect this property, which we think is rather unlikely.

Page 15, lines 264-265: The authors state that the waves are well-defined for a ratio > 13.7. Could you please provide a reference for this threshold? Or is this an observation made from the present analysis?

We realized that this statement is badly phrased. We simply meant to state the mean ratio for the considered wave events. We removed "well-defined" from the sentence so that it reads: "On average the waves have a mean eigenvalue ratio  $\lambda_{med}/\lambda_{min}$  of 13.7."

Page 17, point 1: "Influences of extreme solar wind conditions can be seen in occasional sudden increases of the time between observation of two events." In my opinion, the present study does not provide sufficiently convincing evidence of the influence of extreme solar wind conditions to support this statement. I would suggest to tone it down to be more in line with the findings of the present work ("may be seen" instead of "can be seen" for example). Also, as it is now, it contradicts what is stated in the abstract (at lines 8-9). We rephrased the sentences as suggested.

Figure 15 and associated text: I am not sure I understand how the parameters displayed in Figure 15 are obtained. Did the authors run their model for all steepened wave observations in their data set? And similarly, did they calculate the associated values for the resistivity and viscosity based on Eq 25 and 26 for each interval, using plasma observations made simultaneously with the detection of steepened magnetic field structures?

For each of the wave events we used the values for skewness and amplitude obtained through fitting as input for Eq. (21) and Eq. (24), which were deduced from simulations. For the reference values we used plasma observations made simultaneously to the corresponding steepened magnetic field structure. We have added the following to the manuscript: "The reference values  $\eta'_{ros}$  and  $\nu'_{ros}$  were computed using Eq. (31) and Eq. (30) and the viscosity  $\nu_{sim}$  and resistivity  $\eta_{sim}$  approximations were computed using Eq. (22) and Eq. (25).", in which we explicitly refer to the equations we used to compute the values. Please note that the numbering of the equations changed, because we added one equation in section 3. Hence, Eq. (21) is now Eq. (22) and so on.

Lines 508-509: "This change of the interaction region is most likely caused by transient solar wind events, which is supported by the observation of a smooth simultaneous increase of the mean magnetic field." According to Figure 5, an increase of the background magnetic field is observed during thoses intervals, rather than the "smooth increase" described here, which reads as if the field strength changes progressively over the course of the event. I would suggest to reformulate this. It could also be interesting to add to the discussion that measurements from SREM could provide additional information regarding the solar wind conditions, and Enlil simulations could show whether transient solar events may be reaching the comet at these times (see for example the study by Witasse et al., 2017, "Interplanetary coronal mass ejection observed at STEREO-A, Mars, comet 67P/Churyumov-Gerasimenko, Saturn, and New Horizons en-route to Pluto. Comparison of its Forbush decreases at 1.4, 3.1 and 9.9 AU", doi:10.1002/2017JA023884). This additional analysis lies of course beyond the scope of the present study, but it'd be worth mentioning.

We reformulated the mentioned sentence and added the following information about SREM and Enlil simulations: "For example Rosetta-SREM (Standard Radiation Environment Monitor) measurements could provide additional information regarding solar wind conditions and ENLIL simulations (Witasse et al. 2017) could be used to determine if transient solar wind events reach the comet at certain times.". Lines 514-515: "The pattern the minimum variance direction exhibits resembles the general ion motion close to the nucleus." It would be interesting to discuss what are the possible implications of this finding regarding the source of the waves. Would this hint at one of the instabilities mentioned in previous studies?

Ring-beam instabilities are likely not possible in this region since there is not enough space to form a full ring distribution and the ions move mostly radially away from the comet. However, even if this pattern hinted at one generation mechanism we cannot be sure if this pattern is a consequence of the generation mechanism or if it is a consequence of the interaction between the ambient plasma and the waves. Without more information about the propagation orientation we think discussions about the ion velocity pattern and a possible connection to a generation mechanism are highly speculative.

Lines 519-520: "While the skewness increases with rising neutral gas density, the amplitude decreases" Would it be possible to distinguish between the increase in neutral gas density due to cometary activity and that due to the distance from the comet? If so, could this help in identify the source region of the waves, assuming that the skewness increases as the waves evolve and steepen with time?

With only single-point observations it is difficult to differentiate between those two effects, as we have no information how the properties of the structures evolve in the inner coma. In general, steepening occurs within one "wavelength", hence compared to the length scale of the cometary interaction region steepening occurs quickly. In order to deduce information about the source region from the skewness accurate information about diffusive mechanism or in general about all other possible influences on a waves skewness is needed. With the information available we do not think that this is possible.

**Technical corrections:**

Page 6, line 115: change "implies" to "is associated with" ("implies" suggests that there's a direct causation, which cannot be established on the sole basis of the present study)

We replaced "implies" with "is associated with".

Figure 10: The right-hand side of the right panel seems to be cut: there's a few bins missing to reach 180 degrees, with only a vertical line remaining around 170 degrees.

The figure was corrected.

Page 15, lines 273-274: "the following analysis was only performed for the periods in which diamagnetic cavities were available to adjust the offsets." ! "the following analysis was only performed for the periods in which observations of the diamagnetic cavity were available to adjust the offsets." Changed as suggested.

Page 17, point 4: "at an angle to the Sun" -¿ to the Sun-comet line? to the Sun-spacecraft line?

We added "to the Sun-comet line".

*Line 418: "es" -¿ "as"* Corrected.

Equation 22: "s" should be "S" Corrected.

Line 471: "discrepancy" reads as if this difference between the two parameters is an error, whereas it is actually an observation that is made here, based on the model, unless I am mistaken. I would suggest to reformulate this sentence, for example  $\eta$  and  $\nu$  differ by a factor of ~ 1000" We have rephrased the sentence as suggested.

Line 483: "to low" -¿ "too low" Corrected.

"During this period occasional transitions into regions free of wave events within the span of 1 - 2 days were observed." This sentence reads as if Rosetta was probing a different part of the cometary environment, which didn't have such steepened waves, during these intervals. However, based on the presented analysis (and the next sentence) it is rather that the waves "disappear" from the cometary environment during these intervals. I would suggest to reformulate this sentence to better convey this. We reformulated the sentence as follows: "During this period occasional intervals over 1 - 2 days, in which these waves vanish, were observed."

Line 515: Again, the angles "to the comet" and "to the Sun" should be rather to the "comet-spacecraft line" and to the "Sun-comet" or "Sun-spacecraft" line to be unambiguously defined.

We have added "spacecraft-comet line" and "Sun-comet line".

---

## Referee Report (RR1)

I thank the authors for the extensive answers they have given to the comments by (both) referees.

After reading through the revised paper, there are still a few things that need to be cleared up.

- I would completely take out the word "event" it is too confusing, with all the mixed versions of describing the steepened waves.
  Maybe it would be good to use an abbreviation, stw, for steepened waves? It gets very confusing with "steepened wave", "wave event", "wave", etc.
- What I am missing is what the steepened waves look like in all three components of the magnetic field. This is also not present in the first paper, where the events were sought. Here is an example (from AMDA) of the waves shown in Fig. 5 (top left). It gives the reader at least an impression of what these steepened waves look like in the components.

[Figure]

- Line 219: The authors give here as one explanation for the tableau in Fig. 3a that "the observed stagnation may be caused by underestimating the local mass-loading". I do not quite see how this can be a solution. Do the authors mean that adding the mass loading of the other masses will completely re-arrange all bars in the histogram, and thereby removing the plateau?
- Line 232: "sharp increases coincide" → "sharp increases in the inter-wave time coincide"
- Line 236: "Shortly before" here I think the authors are using a euphemism because "shortly" here is 40(!) and 16 hours.
- Line 244: "a time span of multiple hours" this should then be "a time span of dozens of hours" regarding the previous point.
- Line 256: "However, adjacent to HCSs are very high plasma densities" I do not understand why the high plasma density is "adjacent" of the HCS, while the latter should have the highest density, or am I missing something?

- Line 302: "unfiltered data" -- is "unfiltered" just hi-res data? filtering the data has not been mentioned before.
- Line 309: "For the following analysis …" Start a new paragraph here. And add a short sentence that here "the individual steepened waves are fitted" and then that you only use fits with an R greater then 0.7
- Line 321: "the footpoin in the magnetic field data" How do the authors determine the "footpoint" for example in the cases with "whistler waves" present such as in Fig. 7b?
- Line 330: "the waves are highly non-linear" I would say "can be", because 0.4 is not really "highly non-linear".
- Line 348: Why is there an upper limit for the eigenvalue ratio of 40?
- Line 409: The compressional nature of these waves – and the mainly strongly oblique propagation direction
-

---

## Author Response (AR2)

**Response to review by Martin Volwerk (Referee #2)**

We wish to thank Martin Volwerk for his valuable input and evaluation of our manuscript. Below, we have included the referee comments in italics and our own response in regular text.

*I would completely take out the word "event" it is too confusing, with all the mixed versions of describing the steepened waves. Maybe it would be good to use an abbreviation, stw, for steepened waves? It gets very confusing with "steepened wave", "wave event", "wave", etc.*
We have removed all the occurrences of "wave event" and refer to the structures as steepened waves or waves consistently throughout the manuscript.

*What I am missing is what the steepened waves look like in all three components of the magnetic field. This is also not present in the first paper, where the events were sought. Here is an example (from AMDA) of the waves shown in Fig. 5 (top left). It gives the reader at least an impression of what these steepened waves look like in the components.*
Figure 1 of the updated manuscript contains magnetic field components for two different intervals (16 July 2015 and 21 November 2015) with multiple occurrences of steepened waves.

*Line 219: The authors give here as one explanation for the tableau in Fig. 3a that "the observed stagnation may be caused by underestimating the local mass-loading". I do not quite see how this can be a solution. Do the authors mean that adding the mass loading of the other masses will completely rearrange all bars in the histogram, and thereby removing the plateau?*
While the period considered in the study is heavily water dominated other neutral gas components are still present. $CO_2$ for example has a higher

molecular mass than $H_2O$. Hence, when the contribution of other neutral gas components is not taken into account, the local mass-loading may be underestimated. Since the outgassing is anisotropic the contributions of the other masses will change with time an space. This may cause a reordering of the bars in the histogram.

*Line 232: "sharp increases coincide" -¿ "sharp increases in the inter-wave time coincide"*
Changed as suggested.

*Line 236: "Shortly before" here I think the authors are using a euphemism because "shortly" here is 40(!) and 16 hours.*
We have adjusted the sentence as following: "A day before, on 13 September 2015, the interaction region is completely different."'

*Line 244: "a time span of multiple hours" this should then be "a time span of dozens of hours" regarding the previous point.*
Changed as suggested.

*Line 256: "However, adjacent to HCSs are very high plasma densities" I do not understand why the high plasma density is "adjacent" of the HCS, while the latter should have the highest density, or am I missing something?*
HCS are essentially just a reversal in the magnetic field direction. The high plasma densities, which are also known as heliospheric plasma sheets, typically surround the HCS and are, hence, adjacent to the HCS. More information about this can be found e. g. in [Tsurutani et al. 2016, Heliospheric plasma sheet (HPS) impingement onto the magnetosphere as a cause of relativistic electron dropouts (REDs) via coherent EMIC wave scattering with possible consequences for climate change mechanisms, J. Geophys. Res. Space Physics, 121, 10,130– 10,156, doi:10.1002/2016JA022499] (Figure 1) or in [Lavraud et al. 2020, The Heliospheric Current Sheet and Plasma Sheet during Parker Solar Probe's First Orbit, American Astronomical Society, 894, L19, 10.3847/2041-8213/ab8d2d].

*Line 302: "unfiltered data" – is "unfiltered" just hi-res data? filtering the data has not been mentioned before.*
Depending on which data product from the PSA is used the magnetic field data has been filtered to e.g. remove influences of reaction wheels or as a

consequence of resampling. The unfiltered refers to data which explicitly has not been filtered since a filter can affect the shape of the steepened wave depending on the chosen filter parameters.

*Line 309: "For the following analysis . . . " Start a new paragraph here. And add a short sentence that here "the individual steepened waves are fitted" and then that you only use fits with an R greater then 0.7*
As suggested we have added the following lines to the manuscript: "For the following analysis the individual steepened waves are fitted and only fits with an adjusted R-squared value above 0.7 are further analyzed."

*Line 321: "the footpoint in the magnetic field data" How do the authors determine the "footpoint" for example in the cases with "whistler waves" present such as in Fig. 7b?*
The whistler waves superimpose the magnetic field in front of the steep edge. Hence, we assume that the whistler waves approximately oscillate around the "footpoint" of the steepened wave and determine it accordingly. In the rare cases where the whistler waves are significantly more developed than in Fig. 6b, the adjusted R-squared value of the fit will fall below the threshold of 0.7 an the waves are discarded for the following analysis.

*Line 330: "the waves are highly non-linear" I would say "can be", because 0.4 is not really "highly non-linear".*
We have changed "the waves are highly non-linear" to "the waves can be highly non-linear".

*Line 348: Why is there an upper limit for the eigenvalue ratio of 40?*
We imposed an upper limit on the eigenvalue ratio because disturbances caused by the components on the spacecraft, like heaters, typically have a high eigenvalue ratio ($> 40$) and we wanted to exclude those. To make sure the upper limit does not skew the results we computed the minimum variance direction again without the upper constraint. Figure 1 was computed analogously to Fig. 10 in the manuscript, just without the upper constraint on the eigenvalue ratio. The distribution of the angles only changes marginally.

*Line 409: The compressional nature of these waves – and the mainly strongly oblique propagation direction*
We have changed the sentence as following: "The compressional nature of

[Figure]

[Figure]

Figure 1: Histograms of the angle between the minimum variance direction and the background magnetic field (a) and the spacecraft-Sun connection line (b) without the upper limit for the eigenvalue ratio.

these waves (Engelhardt et al., 2018; Hajra et al., 2018b) and the mainly strongly oblique propagation direction are clear indicators that they behave like fast magnetosonic waves."